# Navigating the Effect of Parametrization for Dimensionality Reduction

**Haiyang Huang**\* **Yingfan Wang**\* **Cynthia Rudin**
Duke University
{hyhuang, yw416, cynthia}@cs.duke.edu

## Abstract

Parametric dimensionality reduction methods have gained prominence for their ability to generalize to unseen datasets, an advantage that traditional approaches typically lack. Despite their growing popularity, there remains a prevalent misconception among practitioners about the equivalence in performance between parametric and non-parametric methods. Here, we show that these methods are not equivalent – parametric methods retain global structure but lose significant local details. To explain this, we provide evidence that parameterized approaches lack the ability to repulse negative pairs, and the choice of loss function also has an impact. Addressing these issues, we developed a new parametric method, ParamRepulsor, that incorporates Hard Negative Mining and a loss function that applies a strong repulsive force. This new method achieves state-of-the-art performance on local structure preservation for parametric methods without sacrificing the fidelity of global structural representation. Our code is available at https://github.com/hyhuang00/ParamRepulsor.

## 1 Introduction

Dimension reduction (DR) methods are incredibly useful for data analysis. They provide a bird's eye view of a dataset that shows clusters and their relationships. These algorithms have been used for examining and processing images [1], text documents [2, 3], and biological datasets [4–8]. The successes of modern DR methods can mostly be attributed to neighborhood embedding (NE), which is the basis for modern DR methods [9] including $t$-SNE, LargeVis, UMAP, and PaCMAP [10–13]. These algorithms aim to optimize the low-dimensional layout of the data, such that the high dimensional local structure (i.e., neighborhoods) are preserved.

A major weakness of existing NE algorithms is that they struggle with adaptability to large, incrementally updated datasets. These algorithms depend on a $K$-Nearest Neighbor graph, encompassing the entire dataset, to generate the embedding. Consequently, the introduction of new data necessitates a complete re-computation of the embedding, leading to significant time and computational resource demands for large datasets. Although recent adaptations have been developed to optimize only the additional data [14, 13], these modifications potentially alter the original algorithm's objective function, thereby compromising the embedding's quality.

Addressing these challenges, recent developments in combining neural networks with NE algorithms have shown promise. These algorithms maintain the same objectives as traditional NE methods but leverage neural networks to optimize the projection of high-dimensional data into lower-dimensional spaces [16–18]. The integration of neural networks allows these NE algorithms to be effectively trained on large datasets and generalize to unseen data. Throughout this paper, we refer to this class of algorithms as parametric algorithms. However, as shown in Fig. 1, despite the similarity in loss

---

\*Equal contribution.

38th Conference on Neural Information Processing Systems (NeurIPS 2024).

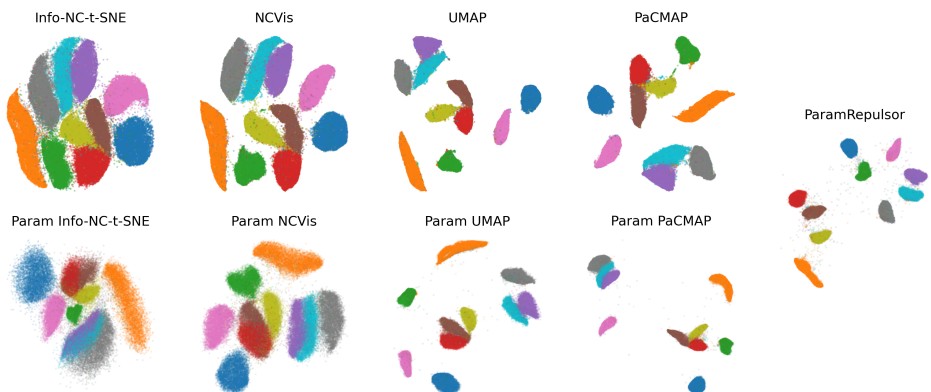

Figure 1: Dimensionality reduction results on the MNIST digit dataset [15]. Parametric methods (bottom row) fail to preserve the local structure of the dataset compared to their non-parametric counterparts (top row). Our method, ParamRepulsor, effectively resolves this problem via Hard Negative Mining.

functions between the non-parametric and parametric versions, their outcomes are often completely different, and such difference has been largely overlooked by machine learning practitioners. This paper aims to illuminate and explain these differences, highlighting that parametrization often leads to worse local structure and visualization. Our investigation reveals that parameterized approaches lack the ability to identify cluster boundaries and separate negatives compared to nonparametric approaches. We further show that DR algorithms using Negative Sampling (NEG)-style loss functions exhibit greater adaptability to parametrization than others using Noise Contrastive Estimation (NCE) or InfoNCE loss. This observation is noteworthy as such discrepancies are not observed in nonparametric approaches.

Building on these insights, we propose a novel parametric DR method that effectively mines hard negatives without relying on labels. Our approach incorporates additional repulsive forces, placing even greater emphasis on pairs we identify as hard negatives. This enhancement ensures better separation and structure preservation, significantly improving the performance of parametric DR. We select a loss function tailored for optimizing the parametric case, addressing local structure preservation. Our new DR algorithm, ParamRepulsor, approaches the performance of leading non-parametric methods while surpassing existing parametric approaches in preserving both local and global structure. It offers a functional mapping from high- to low-dimensional space, ensuring superior scalability, adaptability, and generalization to unseen data.

To summarize, our contributions in this study are:

- We conduct a comprehensive analysis of the impact of parametrization on the performance of DR methods, demonstrating that it may compromise local structure. Our findings attribute this issue to insufficient repulsive forces on negative pairs in the parametric setting. Notably, algorithms employing NEG-style loss functions (e.g., UMAP, PaCMAP) exhibit greater adaptability to parametrization than those using NCE-style loss functions (e.g., InfoNC-t-SNE, NCVis).

- Inspired by contrastive learning, we propose ParamRepulsor, a new method that uses hard negative sampling to improve the handling of negative pairs, combined with a contrastive loss tailored for the parametric setting. ParamRepulsor is a novel, fast algorithm that achieves excellent local structure preservation while maintaining global structure.

## 2 Fundamentals of Neighborhood Embedding Algorithms and Contrastive Learning

We provide essential background on Neighborhood Embedding (NE) methods and notation. We notate the high dimensional data as $X = \mathbf{x}_1 \ldots \mathbf{x}_n \in \mathbb{R}^D$, where $n$ is the number of data points, and $D$ is the dimension. NE algorithms aim to preserve predefined high-dimensional similarities within a low-dimensional embedding to reveal the local and global structure of $X$. Specifically,

NE methods identify a mapping function $f_\theta$ that constructs the corresponding low dimensional embedding $Y = \mathbf{y}_1 \ldots \mathbf{y}_n \in \mathbb{R}^d$, where $\mathbf{y}_i = f_\theta(\mathbf{x}_i)$. We will use $\mathbf{y}_i$ and $f_\theta(\mathbf{x}_i)$ interchangably. For nonparametric DR methods, the function $f_\theta$ is not defined outside of $\mathbf{x}_1, \ldots, \mathbf{x}_n$, though it is possible to interpolate. For visualization purposes, $d$ is usually set to 2 or 3. Since the introduction of t-SNE [10], these algorithms have become widely used due to their ability to identify clusters and manifolds within high-dimensional data. They typically have two stages:

**Similarity Construction Phase.** For all pairs of points $(i, j)$, their high-dimensional similarity, $s_{ij}$, is captured by a similarity function $\Phi(\mathbf{x}_i, \mathbf{x}_j)$ related to their distance. Due to the curse of dimensionality, the Euclidean distance metric fails to accurately represent distances along the data manifold in high-dimensional spaces [19]. A common solution to this issue is to only consider similarities between $K$ nearest neighbors: $s_{ij}$ is set to be non-zero iff $\mathbf{x}_i$ or $\mathbf{x}_j$ are within the $K$ nearest neighbors of each other, where $K$ is a hyperparameter, usually 15-30.

**Embedding Optimization Phase.** After constructing the graph, NE algorithms try to optimize a function $f_\theta$. The objective is encoded by a loss function $\mathcal{L}(\theta)$:

$$\mathcal{L}(\theta) = \mathbb{E}_{ij \in NN} \mathcal{L}_{NN}(\|f_\theta(\mathbf{x}_i) - f_\theta(\mathbf{x}_j)\|_2) + \mathbb{E}_{ik \notin NN} \mathcal{L}_{FP}(\|f_\theta(\mathbf{x}_i) - f_\theta(\mathbf{x}_k)\|_2), \qquad (1)$$

where $\mathcal{L}_{NN}$ denotes the loss for $i, j$ that are similar (among the $K$-nearest neighbors), and $\mathcal{L}_{FP}$ denotes the loss for pairs that are not nearest neighbors in the high-dimensional space. Typically, $\mathcal{L}_{NN}$ decreases when $\|\mathbf{y}_i - \mathbf{y}_j\|_2$ decreases, and $\mathcal{L}_{FP}$ decreases when $\|\mathbf{y}_i - \mathbf{y}_k\|_2$ increases. Their gradients therefore act like forces that *attract* or *repulse* the $NN$ or $FP$ pairs, respectively.

**Relationship to Contrastive Learning.** The similarity of the decomposition to self-supervised contrastive learning has recently been noted by [20, 18]. Specifically, loss functions of major NE algorithms can be considered as cases of Noise Contrastive Estimation (NCE) [21], Info-Noise Contrastive Estimation (InfoNCE) [22] or Negative Sampling (NEG) [23].

Using the framework above, we dive into details of **t-SNE** [10], **NCVis** [24], **UMAP** [12] and **PaCMAP** [13], which are four major recent NE algorithms.

## 2.1 NCE/InfoNCE-based: t-SNE and NCVis

Both NCE- and InfoNCE-based approaches assume the high-dimensional data similarities (the $s_{ij}$'s) follow an underlying data similarity pattern, represented by an unknown distribution $p$. These methods learn a function $f_\theta$ that generates a similar low-dimensional similarity pattern, described by a distribution $q$, aiming to match $p$. $q$ decreases as the pairwise distances in the low-dimensional space increase, though their exact relationship can vary. Since $q$ represents a probability distribution, it must be normalized to ensure all possibilities sum up to 1. The only difference is that NCE uses a logistic loss, whereas InfoNCE uses a cross-entropy loss for the data distribution match.

Approximating $p$ as a Bernoulli distribution [24] with value 1 for NN pairs and 0 for FP pairs. Assuming that for each step in the optimization, we optimize a batch that contains one NN pair and $m$ FP pairs, $q$ should minimize:

$$\mathcal{L}^{NCE} = -\mathbb{E}_{ij \in NN, ik_{c=1\ldots m} \notin NN} \left( \log \frac{q_{ij}}{q_{ij} + \sum_{c=1\ldots m} q_{ik_c}} - m \log \left( 1 - \frac{q_{ij}}{q_{ij} + \sum_{c=1\ldots m} q_{ik_c}} \right) \right)$$
$$(2)$$

$$\mathcal{L}^{InfoNCE} = -\mathbb{E}_{ij \in NN, ik_{1\ldots m} \notin NN} \left( \log q_{ij} - \log \left( q_{ij} + \sum_{c=1\ldots m} q_{ik_c} \right) \right). \qquad (3)$$

t-SNE, the most popular NE algorithm, utilizes a loss defined over the full data set. The raw t-SNE loss is usually written as the KL-divergence between high-dimensional and low-dimensional conditional probability distributions $p$ and $q$. Here, we separate the loss following [25, 13]. [18] notes that the exact values of the $p_{ij}$'s have limited impact and can be treated as binary weights without impacting outcomes. To simplify the calculation and allow for mini-batch stochastic gradient descent, [18] rewrote this loss as an InfoNCE loss [22]. Denoting $d_2(i, j) = \|f_\theta(\mathbf{x}_i) - f_\theta(\mathbf{x}_j)\|_2^2 + 1 = \|\mathbf{y}_i - \mathbf{y}_j\|_2^2 + 1$, the t-SNE loss function can be rewritten as an InfoNCE loss:

$$\mathcal{L}^{t-SNE}(\theta) = -\mathbb{E}_{ij \in NN, ik_{1\ldots m} \notin NN} \left( \log \frac{1}{d_2(i, j)} - \log \left( \frac{1}{d_2(i, j)} + \sum_{c=1\ldots m} \frac{1}{d_2(i, k_c)} \right) \right). \qquad (4)$$

Following [18], we call the mini-batch variant Info-NC-t-SNE. We will use it from now on since the vanilla t-SNE loss requires computing pairwise distances between all points in a dataset, and it is challenging to incorporate that into the mini-batch parametric DR framework.

NCVis [24] uses an NCE [21] loss. We denote the number of negative pairs in a batch as $m$, and set $q_{ij} = \frac{1}{d_2(i,j)}$. The NCVis loss is:

$$\mathcal{L}^{NCVis}(\theta) = -\mathbb{E}_{ij \in NN, ik_{1...m} \notin NN} \left( \log \frac{1}{1 + \sum_{c=1...m} \frac{d_2(i,j)}{d_2(i,k_c)}} - m \log \left( 1 - \frac{1}{1 + \sum_{c=1...m} \frac{d_2(i,j)}{d_2(i,k_c)}} \right) \right).$$
(5)

[18] provides a modern implementation for both algorithms. It also provides parametric versions that adopt multilayer perceptrons (MLP) with [100, 100, 100] hidden neurons and ReLU activation.

## 2.2 NEG-based: UMAP

Negative Sampling (NEG) [23] simplifies the modeling process. Define $q_\theta$ to be a similarity function in the low dimensional space:

$$\mathcal{L}^{NEG}(\theta) = -\mathbb{E}_{ij \in NN} \log \left( \frac{q_{ij}}{1 + q_{ij}} \right) - m \mathbb{E}_{ij \in \mathcal{E}} \log \left( \frac{1}{1 + q_{ij}} \right).$$
(6)

UMAP [12] is a DR algorithm that utilizes the NEG loss [18]. Its loss function is

$$\mathcal{L}^{UMAP}(\theta) = -\mathbb{E}_{ij \in NN} \log \left( \frac{q_\theta^{UMAP}(i,j)}{1 + q_\theta^{UMAP}(i,j)} \right) - m \mathbb{E}_{ij \notin NN} \log \left( \frac{1}{1 + q_\theta^{UMAP}(i,j)} \right).$$
(7)

which is NEG with the similarity kernel $q_\theta^{UMAP}(i,j) = \frac{1}{d_2(i,j)-1}$.

## 2.3 PaCMAP

PaCMAP [13] is another recent DR algorithm that achieves high-quality data visualization. Compared to other NE algorithms, PaCMAP's loss function is designed to follow several mathematical design principles, but does not have a probabilistic explanation. The loss function (omitting the mid-near pairs term, as it is only relevant during the initial epochs, see Appendix D) is defined as follows:

$$\mathcal{L}_{ij \in NN} = W_{NN} \frac{d_2(i,j)}{d_2(i,j) + C_1}, \ \mathcal{L}_{ik \in FP} = W_{FP} \frac{1}{d_2(i,j) + C_2}$$

in which the $W$ weights change based on the epoch, and $C_1$ and $C_2$ are set to 10 and 1, respectively.

To study the effect of parametrization on DR algorithms, we extend the PaCMAP framework to incorporate an MLP to map the high-dimensional input to the low-dimensional embedding. We refer to the resulting parametric algorithm as *ParamPaCMAP*. Implementation details are in Appendix F..

# 3 Effect of Parametrization on DR Results

Machine learning practitioners have long believed that parametric NE algorithms behave similarly to their non-parametric counterparts [17, 18]. In this section, we investigate the performance of the aforementioned parametric and non-parametric versions of these algorithms. To understand the effect of parametrization more thoroughly, for each parametric DR method, we additionally implemented three new versions, using a neural network with 0 hidden layers (i.e., a linear model), 1 hidden layer, or 2 hidden layers as a projector. We fix the number of neurons for each layer to 100.

**Observation 1: Parametric NE algorithms typically lead to worse visual effects as well as worse local structure preservation.** Our results indicate that parametric NE algorithms often fail to produce embeddings of the same quality as their non-parametric counterparts, even on simple datasets such as MNIST [15]. The non-parametric methods in the rightmost column of Fig. 2 are able to separate the clusters fairly well, but from the first four columns of Fig. 2, we see that all four parametric algorithms generate clusters that are densely packed with indistinct boundaries, despite the fact that clusters in MNIST are actually separated. These blurred boundaries result in poorer preservation of local structure, with the possible exception of Parametric PaCMAP.

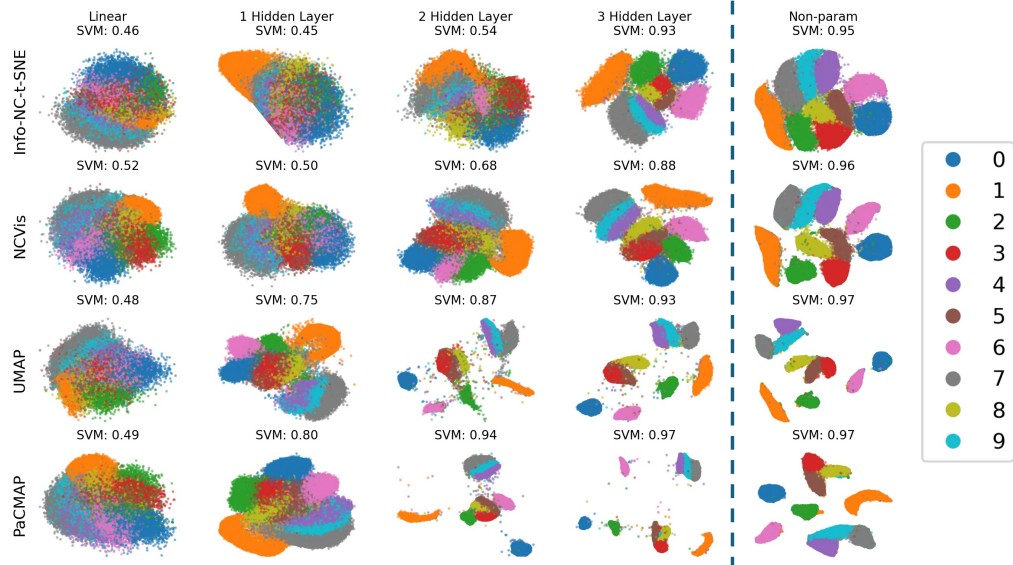

Figure 2: Embeddings of the MNIST [15] dataset generated by various DR methods with different numbers of hidden layers: 0 (Linear), 1, 2, or 3, or non-parametric variant. See Section 5.1 for details of SVM Acc. It is helpful to envision these images in black and white (without labels) to see when clusters would be difficult to visually separate. More datasets/methods can be found in App. C.

The challenge of accurately preserving local structure is exacerbated in scenarios where ground truth labels are unknown, especially in large-scale biological and chemical data, where dimensionality reduction is widely used for data exploration. In these scenarios, users may struggle to identify potential clusters within the large, indistinct conglomerates produced by the NE algorithms.

Figure 2 quantitatively evaluates the quality of the embedding via the SVM accuracy, which measures local structure preservation (described in Section 5.1.) Fig. 3 further quantifies the observation. Here, we sample three kinds of pairs from the points with labels "3" and "8" from the embedding, and calculate the pairwise distance for each kind of pair. NN denotes the pairs of points that are 10-nearest neighbors, and FP denotes pairs of points that are uniformly sampled from the population. MN denotes "mid-near" pairs that are further than NNs but still relatively close (detailed in Sec. 4.) For each embedding, we scale the distance with respect to the scale of the embedding, and calculate the ratio of the mean FP distance to NN distance, as well as the mean MN distance to NN distance. Our analysis reveals that, in comparison to the non-parametric methods, all the parametric counterparts (App. H.1) have a smaller FP distance ratio, meaning further pairs are positioned closer together, which explains the blurred boundaries between clusters.

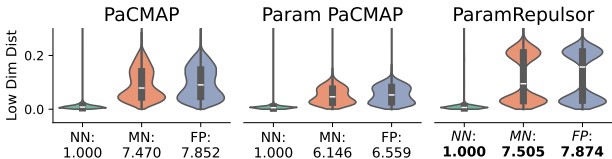

**Observation 2: NE algorithms with NEG loss perform better when parameterized.** A widely accepted explanation for the failure of small neural network projectors, such as those used here, is that they lack the capacity to capture the complexity of the data. While adding more layers to the projector is believed to effectively mitigate the loss in local structure preservation, our experimental results in App. A shows that adding additional layers beyond three yields diminishing returns.

Figure 3: The low-dimensional scaled distance distribution between various types of point pairs with labels "3" and "8" in the embedding of the MNIST digit dataset [15], generated by PaCMAP, ParamPaCMAP, and ParamRepulsor (other methods in App. H.1.) See definitions in Sec. 2 & 4.

As we discuss in App. B, adjusting hyperparameters for parametric DR algorithms—such as the number of nearest neighbors used in NN-graph construction—also had minimal impact on the resulting embeddings. In all cases, the visual quality of the embedding remained suboptimal compared to nonparametric DR methods.

While all four algorithms achieve comparable performance on the MNIST dataset in the non-parametric setting, their ability to adapt to the parametric setting varies significantly. Specifically, NE methods that optimize the NEG loss (UMAP and PaCMAP) perform substantially better than those that optimize the InfoNCE/NCE loss (Info-NC-t-SNE and NCVis). As illustrated in the first four columns of Figure 2, Info-NC-t-SNE and NCVis continue to struggle with local structure preservation in the embeddings when the number of hidden layers is one or two, whereas UMAP and PaCMAP are already capable of grouping similar samples together effectively. Why is this the case? We hypothesize that it is because UMAP and PaCMAP use NEG losses rather than InfoNCE/NCE losses.

We found that PaCMAP's loss is a generalized NEG loss, with a separate similarity function $q_\theta$ defined for $ij \in NN$ and $ij \in FP$. We now state this formally.

**Theorem 3.1.** *The loss of PaCMAP is generalized NEG with low-dimensional similarity functions* $q_\theta^{NN}$ *and* $q_\theta^{FP}$:

$$\mathcal{L}^{PaCMAP}(\theta) = -\mathbb{E}_{ij \in NN} \log \left( \frac{q_\theta^{NN}(\mathbf{y}_i, \mathbf{y}_j)}{1 + q_\theta^{NN}(\mathbf{y}_i, \mathbf{y}_j)} \right) - m\mathbb{E}_{ij \notin NN} \log \left( \frac{1}{1 + q_\theta^{FP}(\mathbf{y}_i, \mathbf{y}_j)} \right), \quad (8)$$

*in which the functions* $q_\theta^{NN}$ *and* $q_\theta^{FP}$ *are*

$$q_\theta^{NN}(\mathbf{y}_i, \mathbf{y}_j) = \frac{\exp(\frac{-C_1}{d_2(i,j)+C_1})}{1 - \exp(\frac{-C_1}{d_2(i,j)+C_1})}, \quad q_\theta^{FP}(\mathbf{y}_i, \mathbf{y}_j) = \frac{1 - \exp(\frac{d_2(i,j)}{d_2(i,j)+C_2})}{\exp(\frac{d_2(i,j)}{d_2(i,j)+C_2})}. \quad (9)$$

Proof: see Appendix D.

To better understand the difference in performance between the NCE, InfoNCE and NEG losses, we compare their terms (Eq 2, 3, 6). The term that attracts the nearest neighbors in these algorithms is consistently in the form of $\log q_{ij}$. This similarity is also evident in the fact that UMAP and t-SNE share the same loss function for nearest neighbors. The key distinction *lies in the treatment of negative pairs* or points that are not nearest neighbors. For NEG-based UMAP, the FP loss for each negative pair $(i, j)$ is $-\log \left( 1 - \frac{1}{d_2(i,j)} \right)$, and for PaCMAP it is $\frac{1}{d_2(i,j)+C_2}$. This term *solely* depends on a negative pair, ensuring that the gradient is large when the negative pair becomes close.

This is in contrast to both NCE and InfoNCE losses, where *each* negative pair term is based on *all* pairs. Theoretical studies [26, 27] in standard contrastive learning have found that such design of loss functions may lead to a reduced gradient and worse performance under a multi-layer perceptron (MLP) model. On the other hand, the NEG loss effectively penalizes the proximity of negative pairs, enhancing the separation between dissimilar points.

## 4   ParamRepulsor

While our ParamPaCMAP algorithm preserves better local structure, the embedding space remains suboptimal, with some clusters that should be distinct still merged together. To solve the problem from its root cause, we propose ParamRepulsor, a novel parametric algorithm built upon our ParamPaCMAP. Pseudocode for ParamRepulsor is found in Alg. 1 and detailed in Alg. 2 in App. F.

There are several major differences of ParamRepulsor from other methods, the major one being the use of **Hard Negative Mining** in the repulsive terms. Our goal is to learn from Hard Negative (HN) Samples – pairs whose DR projections are close but should be far apart. Efficiently sampling HNs could be challenging. Existing approaches either rely on ground truth labels that are not applicable in the unsupervised DR setting [28, 29], or rely on the InfoNCE loss [30] which is less useful for NEG losses. We select *mid-near (MN) pairs* for HN sampling (for the opposite purpose they are used in PaCMAP, where they exert attractive forces). A MN point for point $i$ is identified through the following process: 1) sample $h \sim \text{Uniform}\{1, n\}$ points from the high-dimensional data, and 2) select the second closest point from the sampled set. Here, we use $h = 6$. We justify the use of MN pairs as HN samples based on two key observations.

**Observation 3: Using MN for HN sampling reduces the probability for false negatives.** Existing DR algorithms sample the negative pairs from an (approximately) uniform distribution over all possible pairs. While this approach enhances computational efficiency, it often results in false negatives, which is known to be problematic for contrastive learning [31, 30]. We show that MN pairs are ideal candidates for negatives as they rarely become false negatives.

**Algorithm 1** Simplified Pseudocode for ParamRepulsor

**Require:**
    $\mathbf{X}, n_{NB}, n_{MN}, n_{FP}, n_{\text{epochs}}, f_\theta, \eta, bsz, w_{NB}, w_{MN}, w_{FP}$

1: **Initialize** neural network projector $f_\theta$ with parameter $\theta$
2: **for** $i \leftarrow 1$ **to** $N$ **do**
3:     Sample $n_{NB}$-nearest neighbors, $n_{MN}$ mid-near points.
4: **end for**
5: **for** $epoch \leftarrow 1$ **to** $n_{\text{epochs}}$ **do**
6:     **for** $batch \leftarrow 1$ **to** $n_{batches}$ **do**
7:         Sample $x = x_1 \ldots, x_{bsz}$ from training data, $x^{NN} = NN(x_1 \ldots, x_{bsz})$ from nearest neighbors of each point in $x$, $x^{MN} = MN(x_1 \ldots, x_{bsz})$ from mid-near points (see Sec.4). Sample $x^{FP} = FP(x_1 \ldots, x_{bsz})$ from uniform distribution.
8:         Calculate $y = f_\theta(x), y^{NN} = f_\theta(x^{NN}), y^{MN} = f_\theta(x^{MN}), y^{FP} = f_\theta(x^{FP})$.
9:         $\mathcal{L} = 0$.
10:        **for** $k \leftarrow 1$ **to** $b$ **do**
11:           $\mathcal{L} = \mathcal{L} + w_{NB} \sum_{t=1\ldots n_{NB}} \frac{d_2(y_k, y_k^{NN_t})}{10 + d_2(y_k, y_k^{NN_t})} - w_{MN} \sum_{t=1\ldots n_{MN}} \frac{d_2(y_k, y_k^{MN_t})}{1 + d_2(y_k, y_k^{MN_t})} -$
          $w_{FP} \sum_{t=1\ldots n_{FP}} \frac{d_2(y_k, y_k^{FP_t})}{1 + d_2(y_k, y_k^{FP_t})}$.
12:        **end for**
13:        Calculate gradients $\nabla_\theta \mathcal{L}$.
14:        Update parameters $\theta$ using Adam optimizer.
15:     **end for**
16: **end for**
17: **return** $f_\theta(\mathbf{X})$

**Theorem 4.1.** *The probability that a sampled MN point is a false negative in a dataset of size $n$ converges to 0 at a rate of $O(\frac{1}{n^2})$.*

**Corollary 4.2.** *MN points are less likely to be false negatives than uniformly sampled points in datasets with $n \gtrsim 10^3$. See Appendix E for empirical results and Fig. 21 in Appendix E for projection.*

Proof: see Appendix E. Theorem 4.1 and Corollary 4.2 state that the likelihood for an MN to be a false negative is low. Furthermore, the simplicity of MN sampling ensures efficiency: the sampling cost is still constant for each mid-near point.

**Observation 4: MN pairs are challenging negatives that provide better gradients for local structure preservation.** The shallow parametrization used in NE DR methods ensures that distances in the high-dimensional space remain correlated with those in the low-dimensional embedding. As shown in Fig. 3, in the blurred boundaries of clusters "3" and "8," MN pairs tend to be closer than normal FP pairs in the embeddings of all methods (see Fig. 22 in App. H.1 for other methods). This proximity makes MN pairs challenging negatives for the algorithm, resulting in large gradients during the loss calculation. Fig. 4 illustrates the representations learned by repulsing MN hard negatives. Our approach improves the boundaries between clusters, while maintaining the proximity between close clusters. It not only achieves state-of-the-art cluster separation in parametric methods but also outperforms several non-parametric methods.

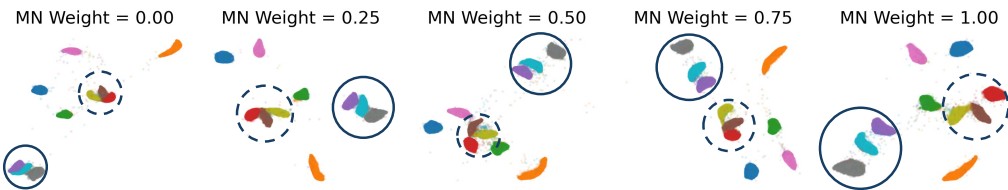

Figure 4: Effect of Hard Negative Mining on MNIST. We progressively increase the coefficient of the *repulsive* force applied to MN hard negatives. Close clusters are circled. Results indicate that Hard Negative Mining alone effectively preserves local structure while maintaining relative proximities.

Besides adopting Hard Negative Mining, we made other technical improvements to further enhance repulsive forces. More details can be found in Appendix F.

# 5   Experiments

Here, we evaluate the performance of our ParamPaCMAP and ParamRespulsor algorithms empirically. To contextualize our findings, we juxtapose our results against those obtained from other contemporary parametric DR algorithms. Visualization for the embeddings generated by all algorithms can be found in App. C.

**Datasets.** We use a wide-ranging collection of datasets across various disciplines. For image analysis, we analyzed the MNIST [15] and Fashion-MNIST (F-MNIST) [32] datasets, along with COIL-20 [33] and COIL-100 [34]. In the domain of computational biology, our assessment leveraged single-cell RNA-sequencing (scRNA-seq) datasets from studies by [35], [36], [37], [38]. Further diversifying our dataset selection, the 20 Newsgroups (20NG) [39] text dataset was included for textual data analysis. The preprocessing of scRNA-seq datasets adhered to the methodology outlined by [40]. Additionally, simulated datasets featuring predefined known structures – such as Circle, Mammoth [41, 42], Gaussian Lineage, and Gaussian Hierarchical [43] – were integrated into our analysis. See Section G.1 for more details. This multifaceted dataset compilation enables a thorough examination of the DR algorithms' performance across a spectrum of datasets.

**Algorithms.** Besides the two algorithms we proposed in this work, ParamPaCMAP (P-PaCMAP) and ParamRepulsor (P-Rep), we also perform experiments on other recent Parametric NE algorithms: Parametric UMAP (P-UMAP) [17], Parametric Info-NC-t-SNE (P-ItSNE) [18], Parametric Neg-t-SNE (P-NtSNE), and Parametric NCVis (P-NCVis) [24, 18]. Besides NE algorithms, we also compare against Geometric Autoencoder (GeoAE) [44], an autoencoder-based DR algorithm. While we note that there are many other parametric DR algorithms, they either aim to serve as an intermediate representation for downstream tasks (i.e., not visualization) [45, 46], or focus only on image dataset only [1]. We refer readers to Section 6 for more details. We compare these algorithms on local and global structure preservation. For each algorithm, we use the hyperparameter settings and the network structure suggested in their implementation. Coincidentally, all the parametric algorithms in our experiment (except for GeoAE) are equipped with a 3-layer 100-neuron fully-connected neural network as their parametric projector $f_\theta$.

**Other setup.** For each experiment, we ran each DR algorithm 10 times using different random seeds to obtain 10 embeddings. We report the average metric measured across these 10 embeddings, highlighting the highest value in **bold**. An independent t-test with a significance level of $p = 0.05$ was conducted to assess significant differences between methods. Metrics not significantly different from the highest value are in *italics*.

## 5.1   Local Structure Evaluation

We first look into the local structure of the embedding, which examines DR algorithms' ability to discover the cluster structure. Following previous works [47, 48, 13, 43], we evaluate local structure using three approaches, with results below. All visualizations can be found in App. C. We achieve state-of-the-art performance in local structure preservation.

**Local Structure 1: $k$-NN Accuracy.** Here, DR is performed and the labels are revealed afterwards. A $k$-NN model then classifies points in the DR projection, with its accuracy as the metric of interest. We perform leave-one-out cross validation, and utilize a $k$-NN classifier to predict the label of the point. For embedding data with good local structure, points that belong to the same class should be close to each other, which would yield a higher $k$-NN accuracy. In this study, we use $k = 10$. Table 1 presents the 10-NN accuracy of each DR algorithm. ParamRepulsor achieves the highest accuracy on 10 out of 14 datasets and comes close to the highest accuracy on the remaining datasets, demonstrating its strong performance in preserving local structure.

**Local Structure 2: SVM Accuracy.** Table 2 in App. H.3 illustrates the SVM accuracy, estimated using 5-fold cross-validation with an SVM classifier. ParamRepulsor achieves the highest accuracy on 9 out of the 14 datasets and achieves near-highest accuracy on the remaining datasets. These results demonstrate that ParamRepulsor attains state-of-the-art performance in preserving local structure.

**Local Structure 3: Nearest Neighbor Kept.** We further evaluate the ability of DR methods to maintain high-dimensional $k$-NN in the low-dimensional space. We use $k = 30$ to provide a more robust estimate of neighborhood preservation ability. Using a larger $k = 30$ value ensures that even if the first nearest neighbor in the high-dimensional space is placed as the tenth nearest neighbor in

Table 1: 10-NN Accuracy of DR methods measured on various datasets. The absence of values indicate the method failed to produce a valid embedding.

| METHOD | P-UMAP | P-ITSNE | P-NTSNE | P-NCVIS | GEOAE | P-PACMAP | P-REP |
|---|---|---|---|---|---|---|---|
| MNIST | *0.965* | 0.830 | 0.862 | 0.829 | 0.791 | *0.968* | **0.969** |
| F-MNIST | *0.733* | 0.714 | 0.714 | 0.626 | *0.718* | *0.744* | **0.778** |
| USPS | *0.957* | *0.939* | *0.940* | *0.938* | 0.846 | *0.960* | **0.957** |
| COIL-20 | *0.843* | – | – | – | 0.724 | *0.853* | **0.887** |
| COIL-100 | 0.145 | – | – | – | 0.611 | 0.896 | **0.928** |
| 20NG | **0.505** | 0.340 | 0.401 | *0.442* | 0.061 | 0.437 | *0.460* |
| KANG | *0.954* | *0.956* | *0.956* | *0.955* | 0.468 | *0.960* | **0.961** |
| KAZER | *0.939* | *0.937* | *0.937* | *0.937* | 0.700 | **0.940** | *0.939* |
| MURARO | *0.960* | *0.961* | *0.961* | *0.961* | 0.565 | *0.961* | **0.962** |
| STUART | *0.851* | *0.854* | *0.853* | *0.854* | 0.394 | *0.855* | **0.856** |
| CIRCLE | *0.901* | *0.900* | *0.904* | **0.911** | *0.898* | *0.904* | *0.895* |
| MAMMOTH | *0.934* | 0.916 | 0.914 | 0.915 | **0.962** | 0.915 | *0.938* |
| LINEAGE | **1.000** | **1.000** | **1.000** | **1.000** | **1.000** | **1.000** | **1.000** |
| HIERARCHY | **1.000** | **1.000** | **1.000** | **1.000** | *0.976* | **1.000** | **1.000** |

the embedding, it is still considered preserved. This approach mitigates the effects of the reduced dimensionality of the embedding, where small shifts can otherwise result in the loss of neighborhood relationships. Table 3 in App. H.3 demonstrates that ParamRepulsor achieves the highest accuracy on 10 out of 14 datasets and nearly the highest on 3 others, showcasing its strong performance in preserving local structure. Additionally, our implementation of ParamPaCMAP performs comparably to the best methods on all but 2 datasets.

## 5.2 Global Structure Evaluation

We evaluate global structure by evaluating the preservation of cluster-level triplet relationships. Cluster-level triplet relationship preservation is important, particularly for computational biologists performing lineage analysis. Following [43], the metric for this is a Spearman (rank correlation). To compute it, we take one cluster centroid $c$ and rank all other centroids based on low-dimensional distance to $c$. We also repeat this for all $C$ cluster centroids and place these rankings in a single vector. We repeat the process for the high-dimensional space and place these rankings in another vector. The Spearman correlation between these two vectors is the result, shown in Table 4 in App. H.3. Out of the 14 datasets, ParamPaCMAP achieves the highest correlation on 5 of them, whereas ParamRepulsor achieves the highest on 4. These results suggest our ideas are powerful for global structure preservation.

## 6 Related Work

The evolution of DR algorithms can be broadly categorized into two distinct phases. In the initial phase, the focus was on the development of methods that preserved only *global structure*. Key techniques in this category include Principal Components Analysis (PCA) [49], Multidimensional Scaling [50], and Non-negative Matrix Factorization [51]. While these methods effectively maintain the global layout of the data, their primary limitation is that they often fail to retain the inherent neighborhoods and clusters of the data.

Subsequent DR methods were developed to address the shortcomings by emphasizing the preservation of *local structure*, specifically focusing on preserving $k$ nearest neighbor relationships in the original dataset. These local methods, such as Isomap [52], Local Linear Embedding (LLE) [53], Laplacian Eigenmap [54], and more recent Neighborhood Embedding (NE) algorithms like t-SNE [10] and UMAP [12], are particularly adept at maintaining cluster structure. However, they may not adequately preserve the overall spatial layout of clusters. NE methods are more frequently used because they show clusters and manifolds in the high-dimensional space that are difficult to see any other way.

NE methods are typically non-parametric, creating a low-dimensional embedding that maps each data point to a location in 2D, but there does not exist a function that maps from the original (high-dimensional) space to the embedding space. To map new points to the low dimensional space, one

typically creates a nonparametric map from high to low dimensions that places new points near their high-dimensional neighbors (assuming one does not want to rerun the algorithm when adding new points). This approach creates crowding problems, where many high-dimensional points map to the same location in low dimensions.

To address the challenges posed by non-parametric NE algorithms, parametric NE algorithms have emerged as an effective solution. These algorithms focus on learning a function that maps data from a high-dimensional space into a low-dimensional embedding, typically using a neural network. Examples of this approach include the Multi-layer Perceptron based Parametric t-SNE [16], DEC [55], kernel t-SNE [56] and Parametric UMAP [17]. Furthermore, recent advancements have integrated concepts from Contrastive Learning and Representation Learning, with significant contributions from TopoAE [57], GeoAE [44], t-SimCNE [1], and Parametric InfoNC-t-SNE [18].

Recently, [13], [1] and [18] discussed the effect of the loss function forces in NE algorithms. Our work differs from them; in our work, we discuss the effect of parametrization, which is not discussed in previous works.

Learning from Hard Negatives has proven effective in supervised learning [28], metric learning [29], as well as contrastive learning [30]. To the best of our knowledge, our work is the first that explores the effect of Hard Negative Mining in dimensionality reduction.

# 7 Discussion and Limitations

Parameterization of DR methods has major practical advantages. It allows for new data to be mapped directly from the high-dimensional space to the low-dimensional space by a function. We introduced a new method called ParamRepulsor, which demonstrates enhanced preservation of local structure without compromising global structure metrics, making it applicable across a broad spectrum of scientific inquiry.

We note that our method also exhibit limitations. Although ParamRepulsor outperforms Parametric UMAP in terms of speed, it requires more computational time than Parametric Info-NC-t-SNE. Other open questions that are not resolved by this work include the design of evaluation metrics that better reflect performance, and choosing the optimal architecture for both preservation and generalization.

# Code and data availability

Implementations of ParamRepulsor/ParamPaCMAP discussed in this paper, along with the code for the experiments, are available at `https://github.com/hyhuang00/ParamRepulsor`. The datasets used in our study are publicly accessible from their original publications.

# Acknowledgement

We acknowledge funding from the National Science Foundation under grants IIS-2130250, IIS-2147061, DGE-2022040 and the National Institutes of Health under grant 5R01-DA054994.

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

# A   Discussion on the Depth of of Neural Network Projector

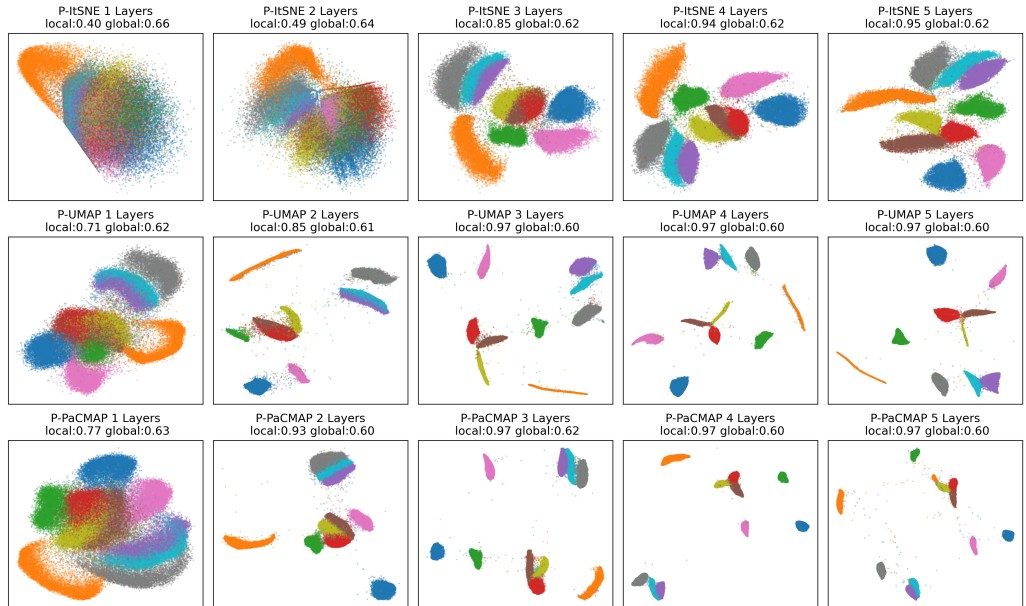

Figure 5: Effect of the number of layers on the MNIST dataset. As a supplement to Fig. 2, we extend the number of layers beyond three for Into-NC-t-SNE, UMAP and PaCMAP. Here, the local metric represents 10-NN accuracy, while the global metric denotes the random triplet preservation. Results show that further increasing the number of layers beyond increasing the number of layers beyond three yields only diminishing and negligible improvements in local structure on all three methods.

Fig. 5 shows the effect of further increasing the number of hidden layers beyond three in the neural network projector for Parametric Info-NC-t-SNE (P-ItSNE) [18], UMAP (P-UMAP) [17], and PaCMAP (P-PaCMAP). P-ItSNE receives little benefits from further increasing layers, while P-UMAP and P-PaCMAP do not receive any further improvements. The magnitude of local structure accuracy increase is rapidly diminishing, and the visual effect is still suboptimal compared to their non-parametric counterpart.

# B Discussion on the Hyperparameter settings of Parametric DR algorithm

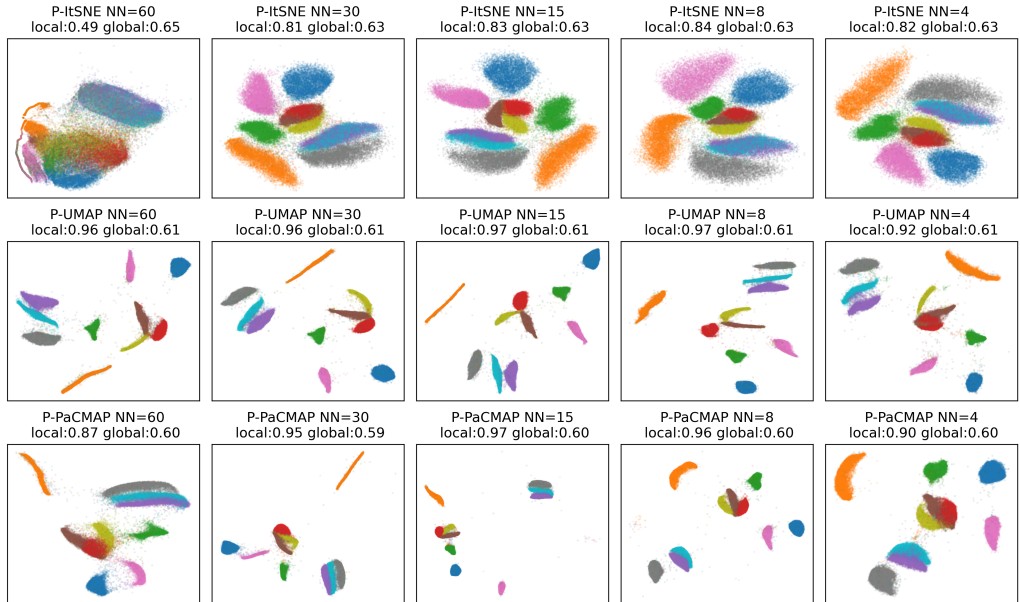

Figure 6: Impact of varying the number of nearest neighbors during NN-graph construction on the MNIST dataset. As in Fig. 5, we evaluate the embeddings using 10-NN accuracy and random triplet preservation to assess local and global structure retention, respectively. The results indicate that, unlike non-parametric algorithms, altering the number of nearest neighbors has minimal effect on the embedding's quality, except for Info-t-SNE, which exhibits structural distortion when NN = 60.

Fig. 6 illustrates the effect of varying the number of nearest neighbors (NN) in P-ItSNE, P-UMAP, and P-PaCMAP. Adjusting the number of NNs during NN-graph construction is commonly regarded as a key mechanism for controlling the local-global structure trade-off in nonparametric DR algorithms [58–60]. Nevertheless, in the parametric setting, modifying the number of NNs had minimal influence on the resulting embeddings. Notably, increasing the number of NNs beyond the typical range (e.g., to 60) can severely disrupt the structure, as observed in the P-ItSNE case.

# C Visualizations from all DR methods

In this section we provide visualizations for the output of all DR methods. Fig. 7 visualizes the output of ParamRepulsor. Notably, ParamRepulsor performs well on all datasets and achieves state-of-the-art on both local and global structure preservation. On MNIST, ParamRepulsor is the only parametric algorithm that separates the clusters with clear boundaries. Compared to nonparametric algorithms, ParamRepulsor has better global structure preservation, as it is able to keep the structure of the mammoth on Mammoth dataset, and keep the structure of the hierarchy on the Hierarchy dataset.

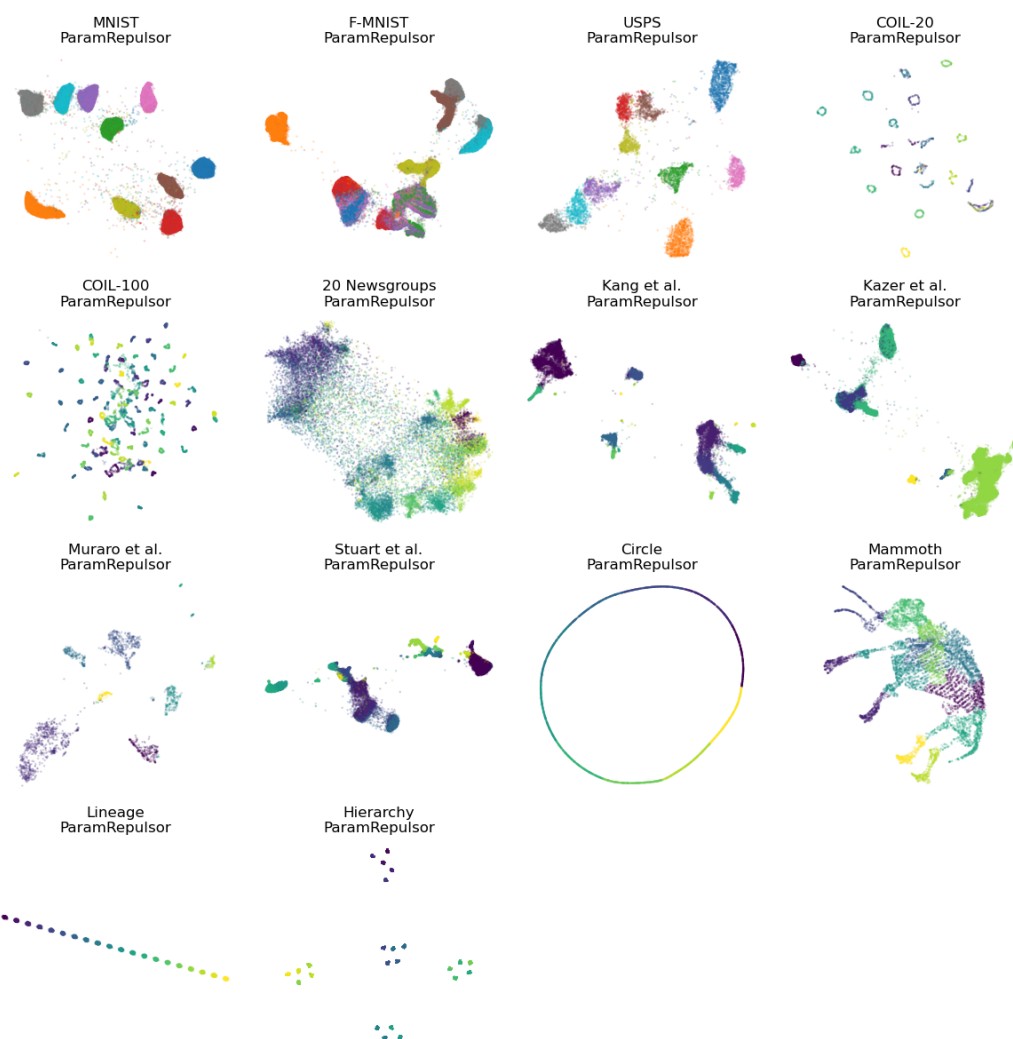

Figure 7: All dimensionality reduction results of ParamRepulsor.

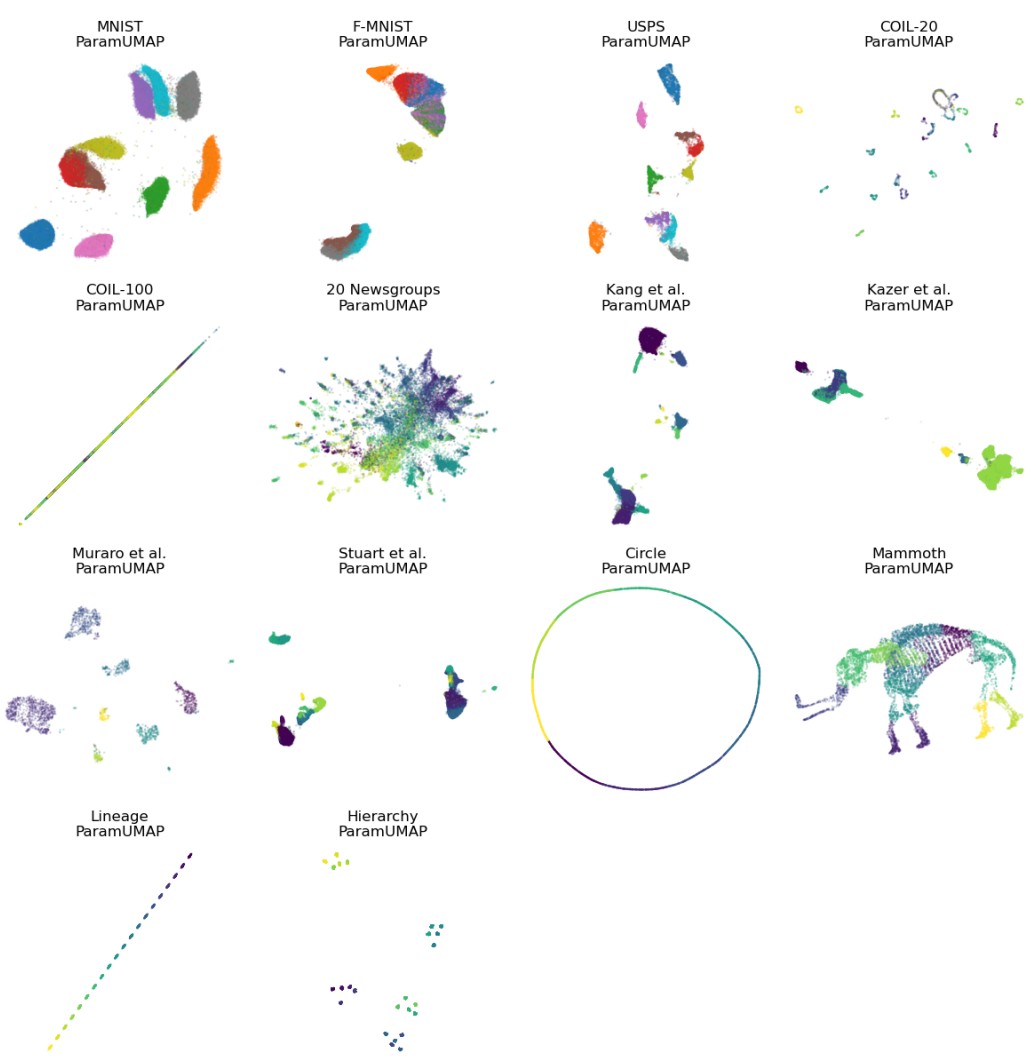

Figure 8: All dimensionality reduction results of ParamUMAP.

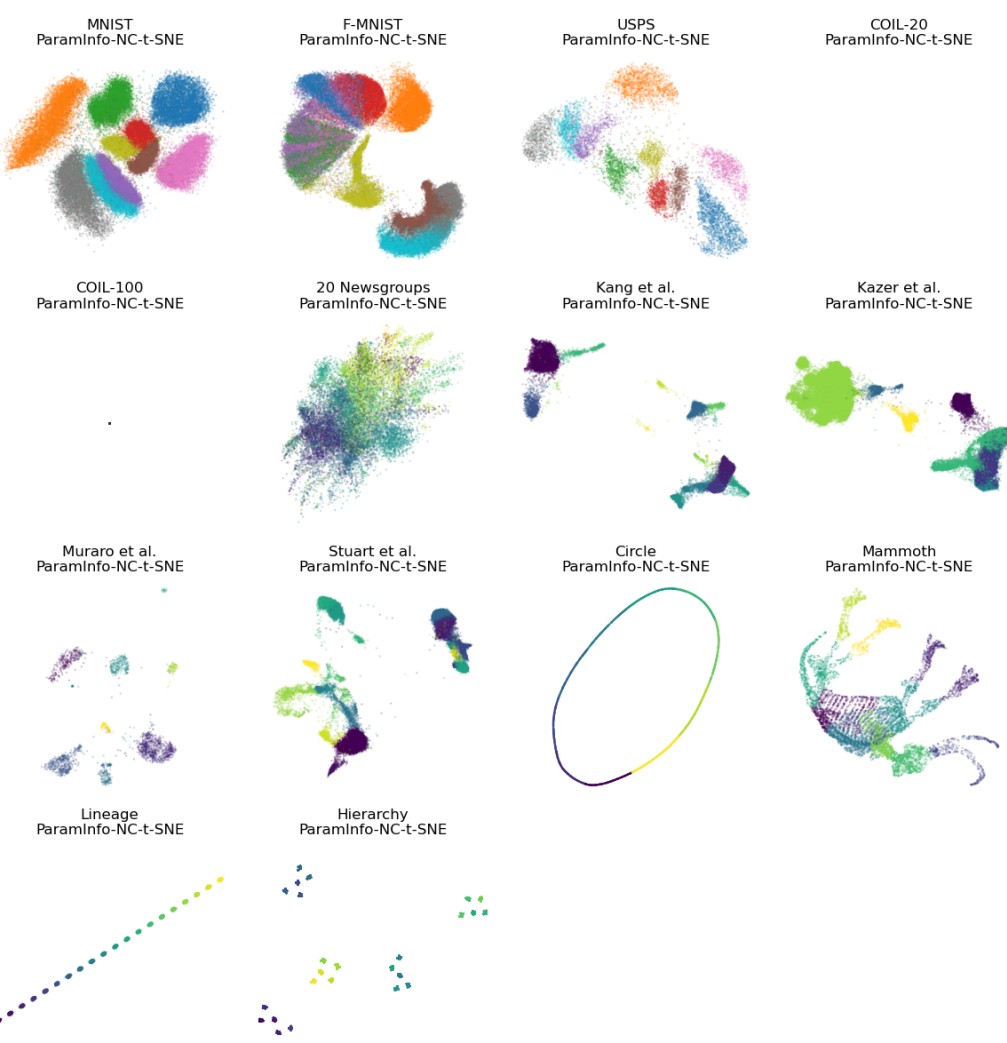

Figure 9: All dimensionality reduction results of ParamInfo-NC-t-SNE.

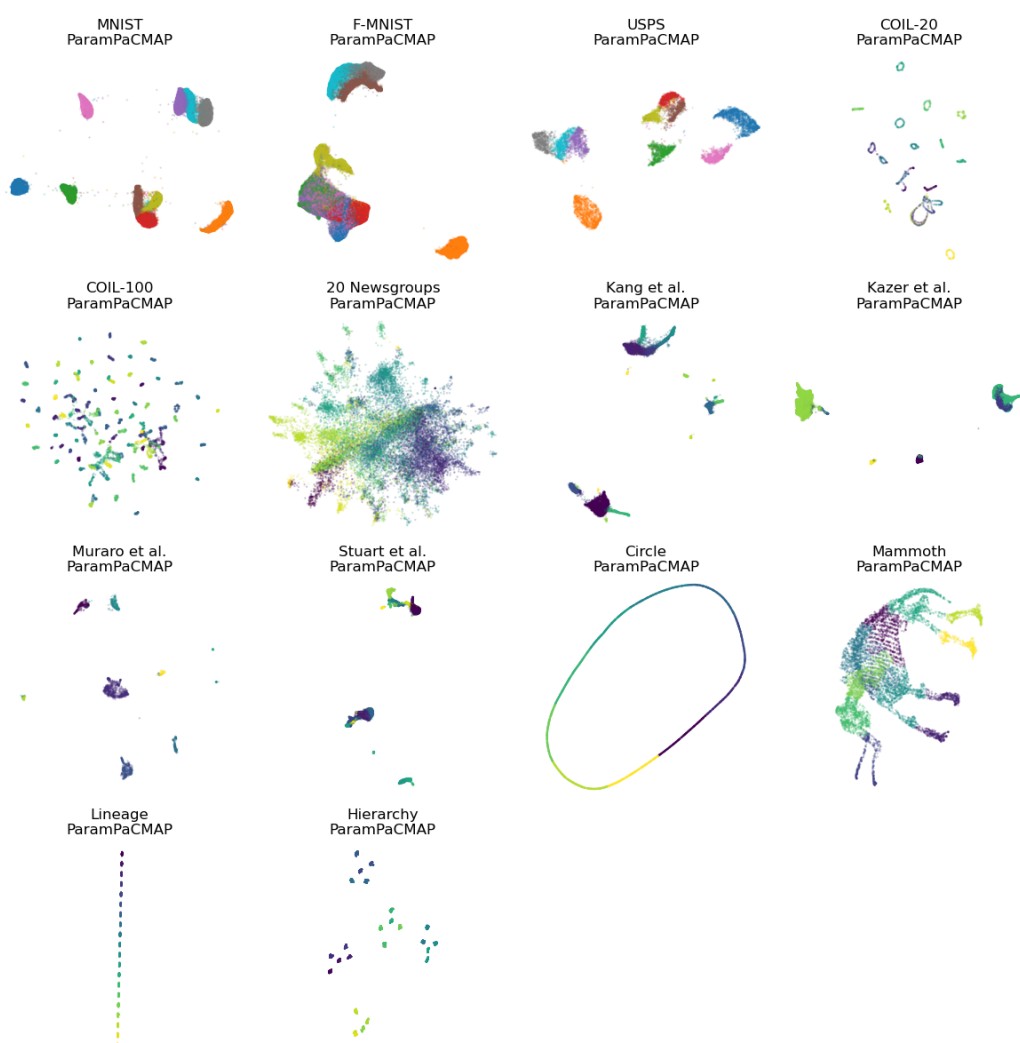

Figure 10: All dimensionality reduction results of ParamPaCMAP.

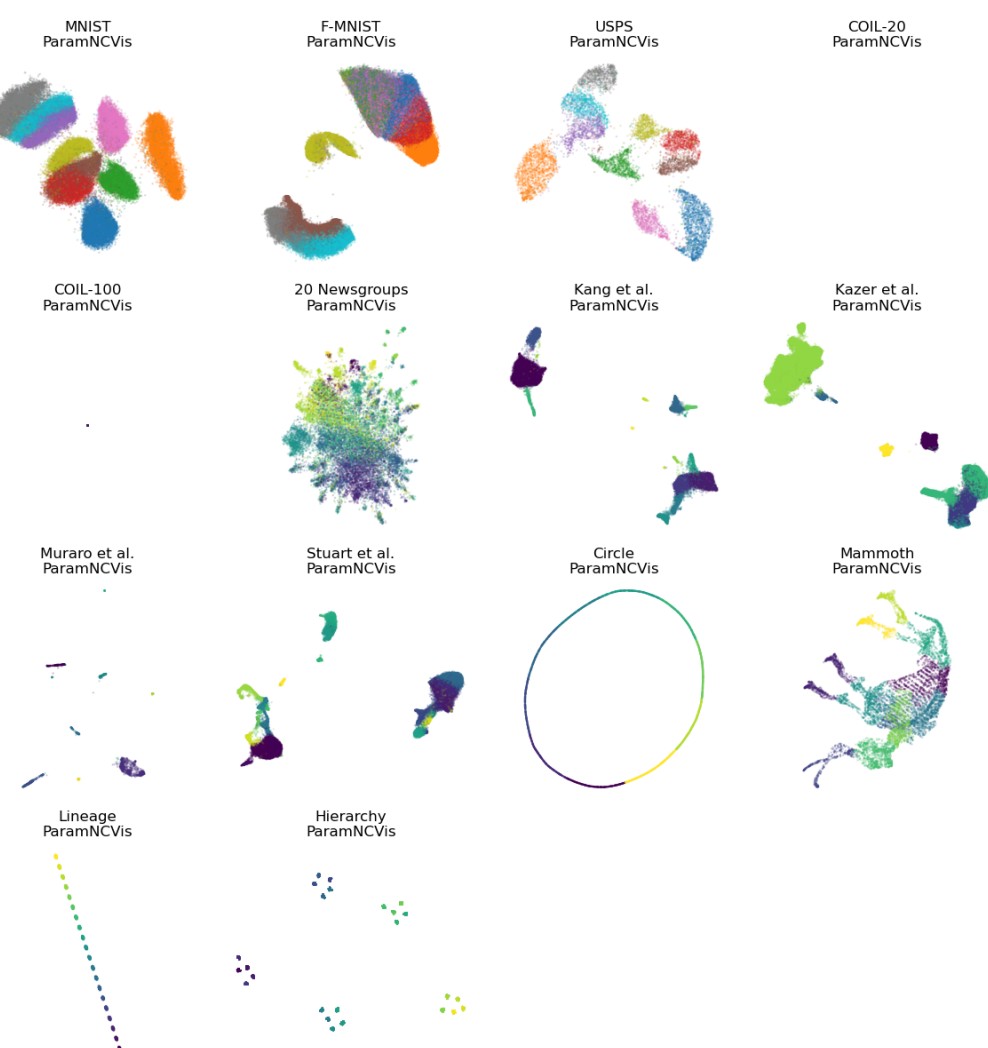

Figure 11: All dimensionality reduction results of ParamNCVis.

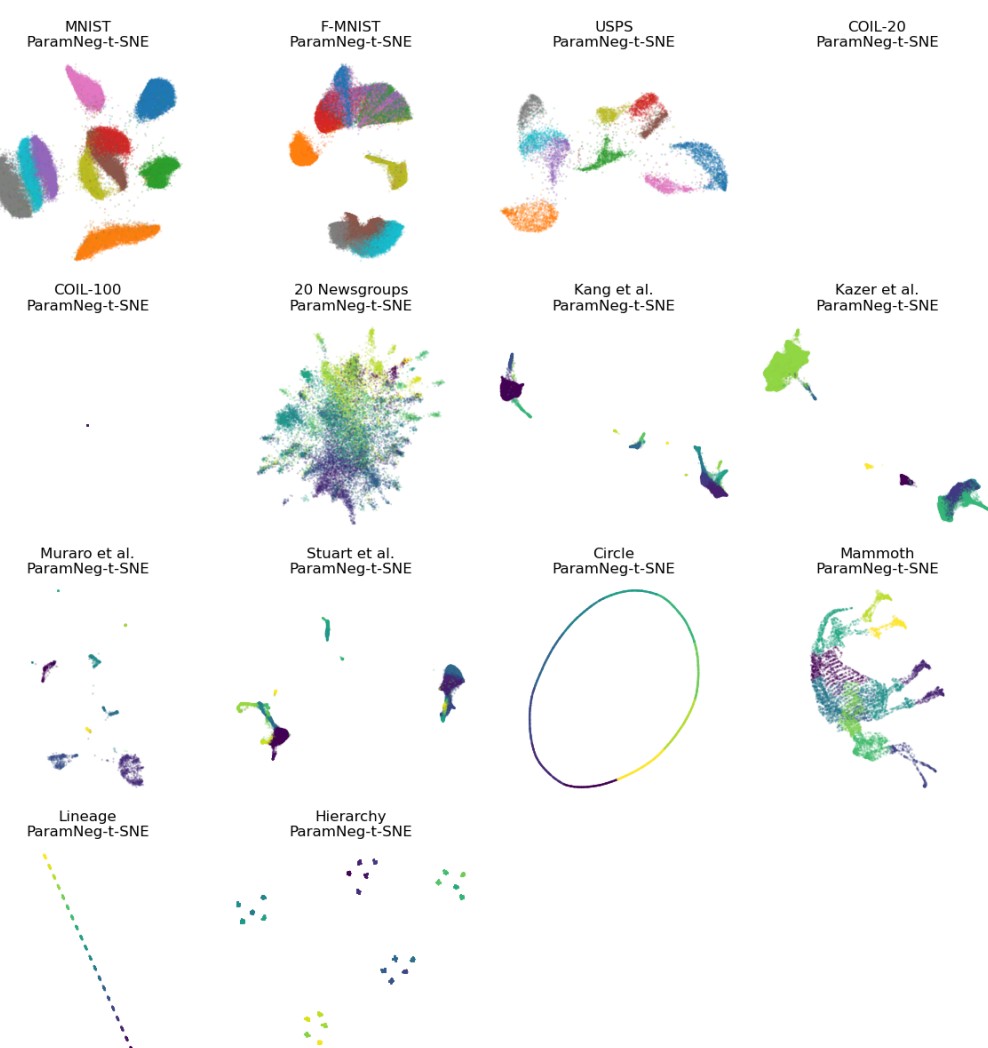

Figure 12: All dimensionality reduction results of ParamNeg-t-SNE.

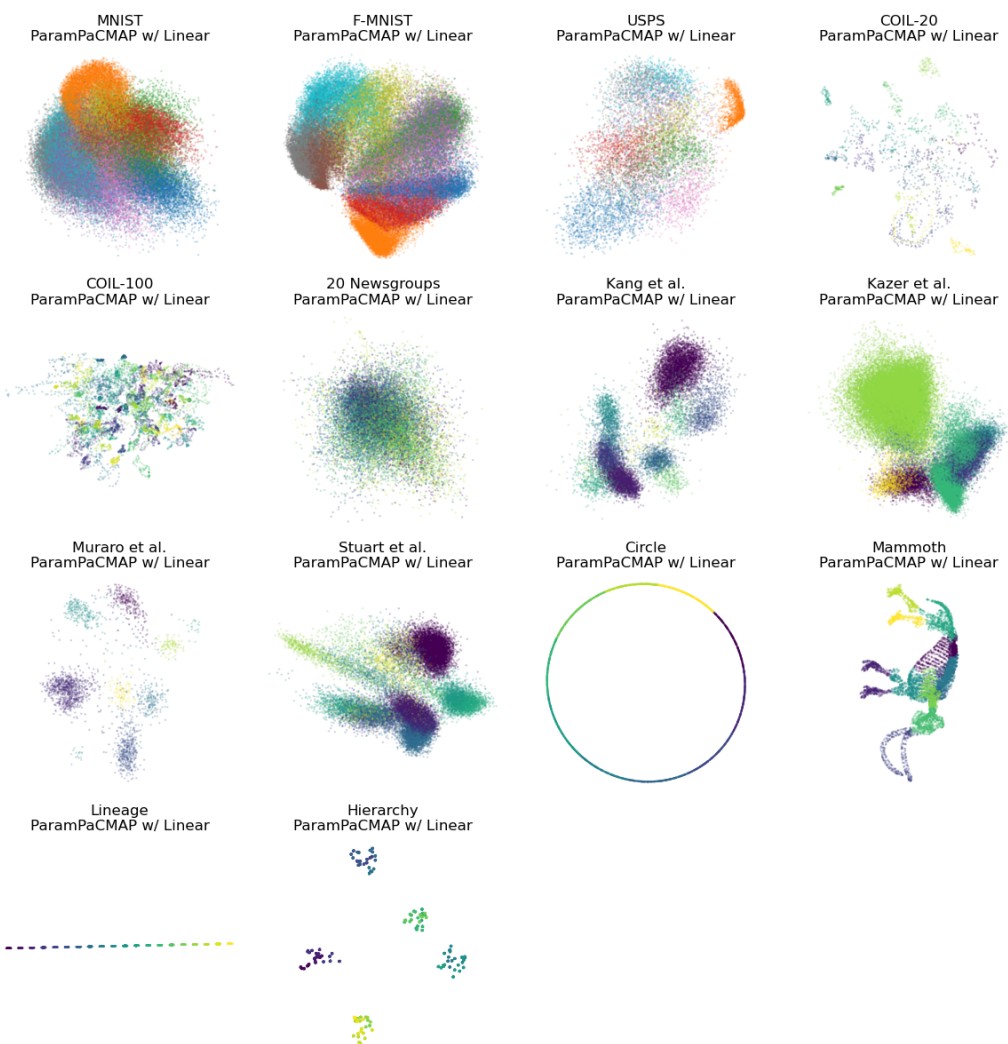

Figure 13: All dimensionality reduction results of ParamPaCMAP with a linear projector.

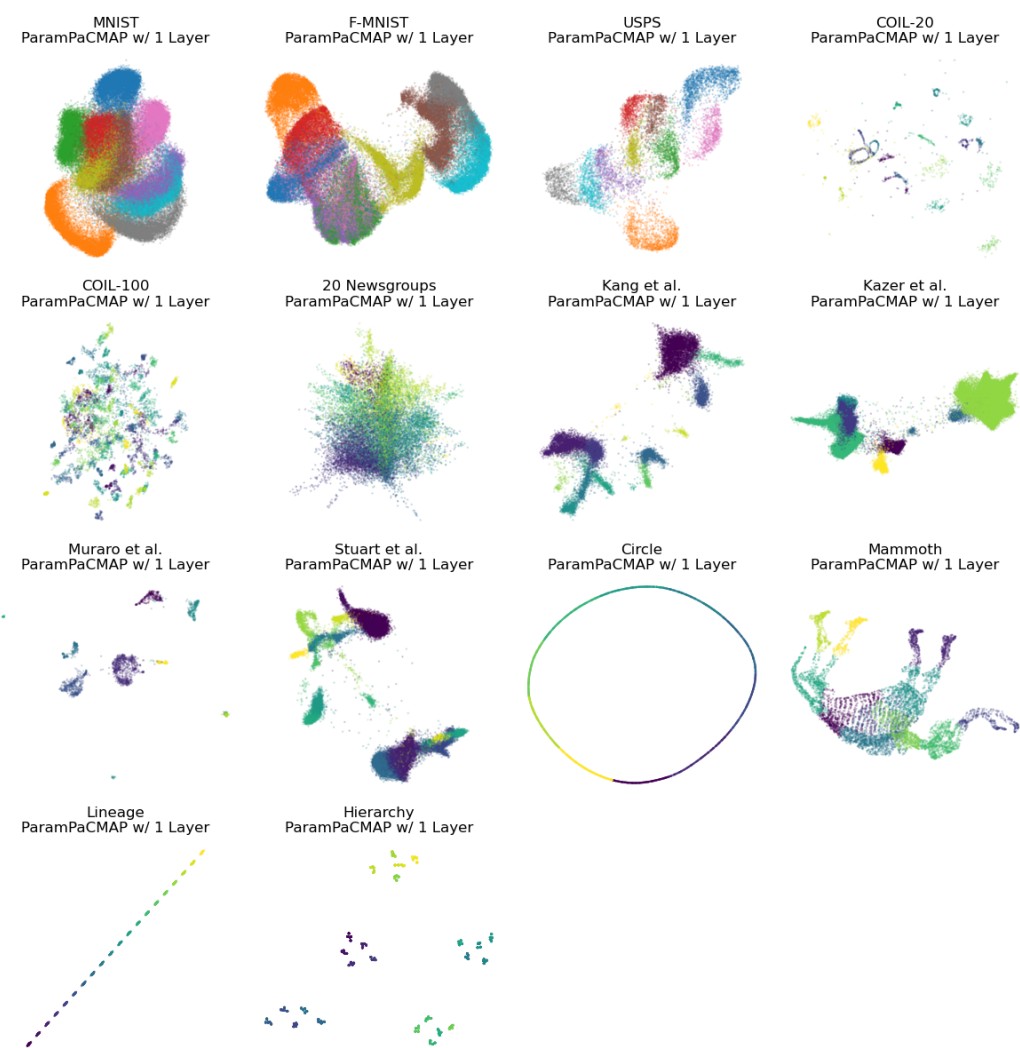

Figure 14: All dimensionality reduction results of ParamPaCMAP with 1 hidden layer.

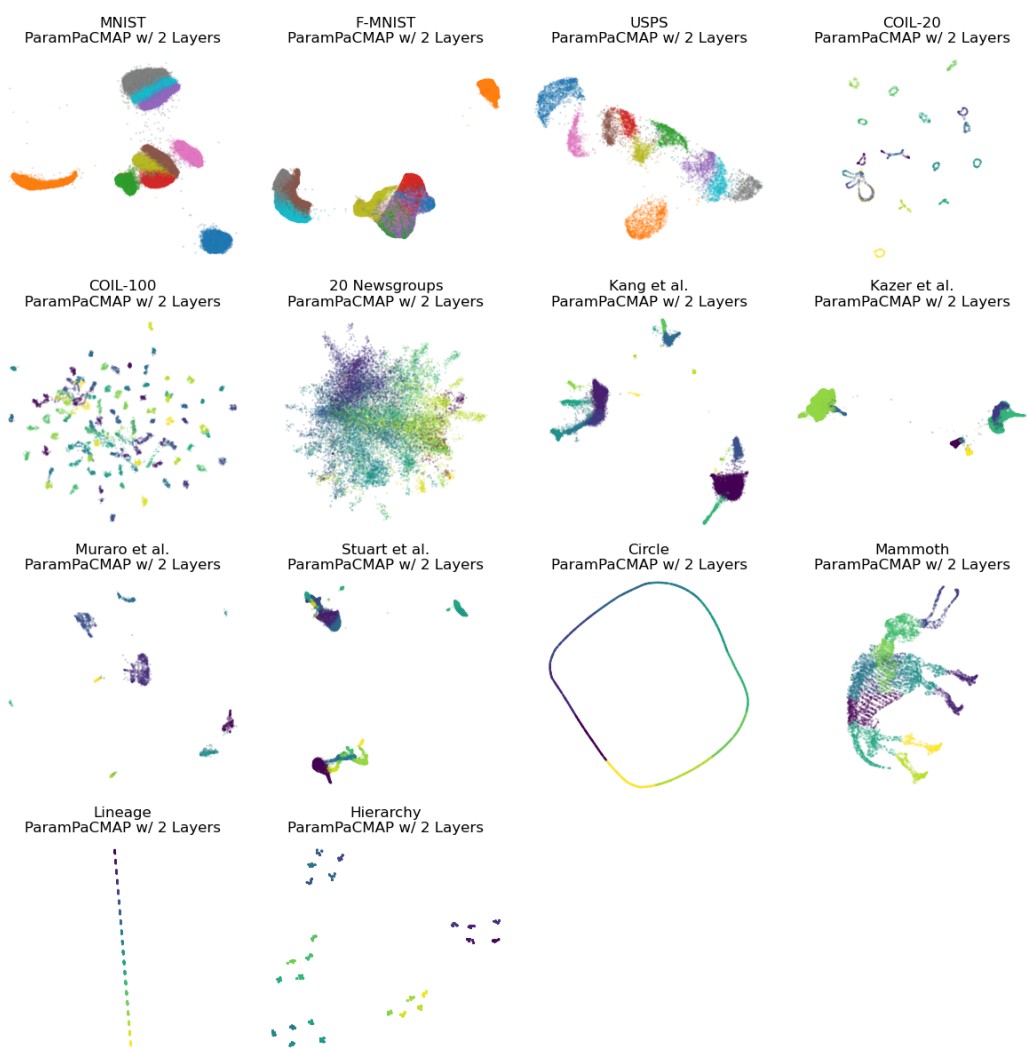

Figure 15: All dimensionality reduction results of ParamPaCMAP with 2 hidden layers.

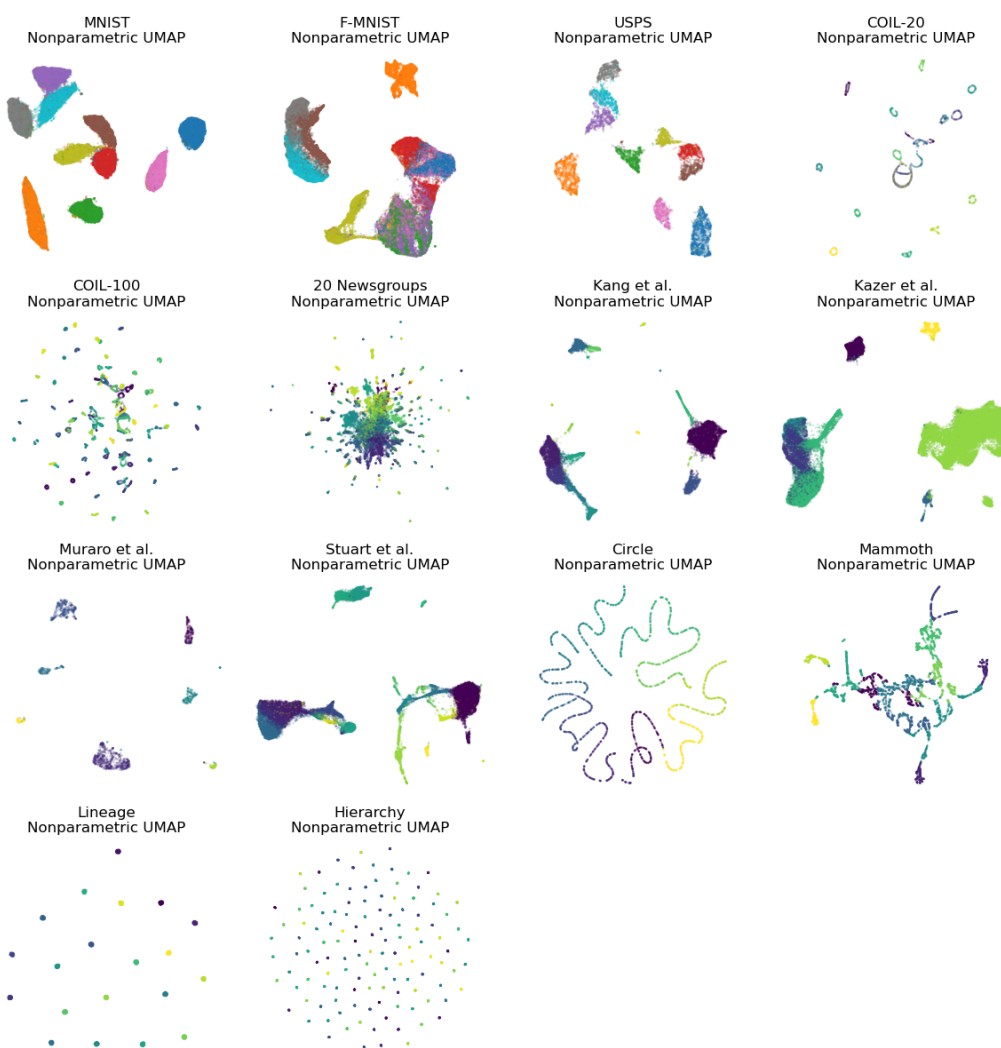

Figure 16: All dimensionality reduction results of nonparametric UMAP.

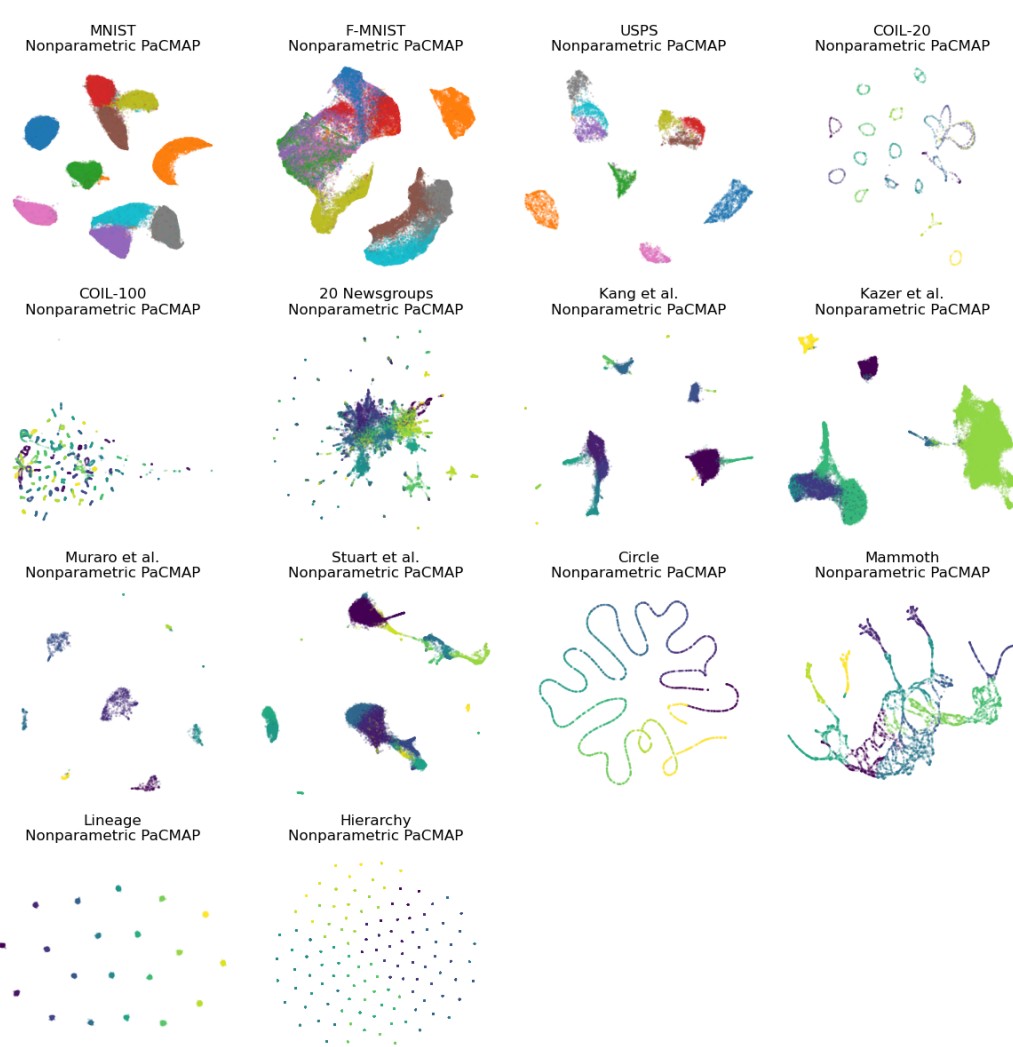

Figure 17: All dimensionality reduction results of nonparametric PaCMAP.

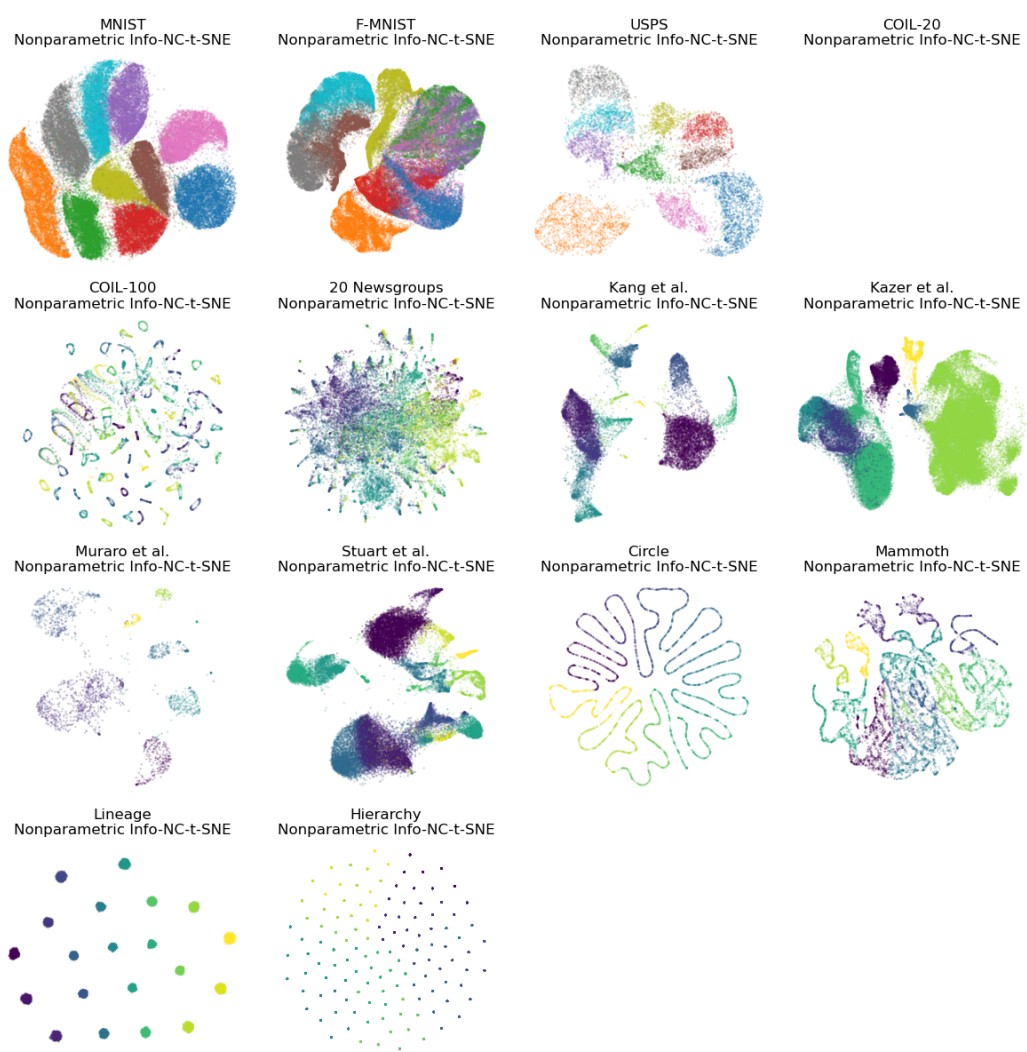

Figure 18: All dimensionality reduction results of nonparametric Info-NC-t-SNE.

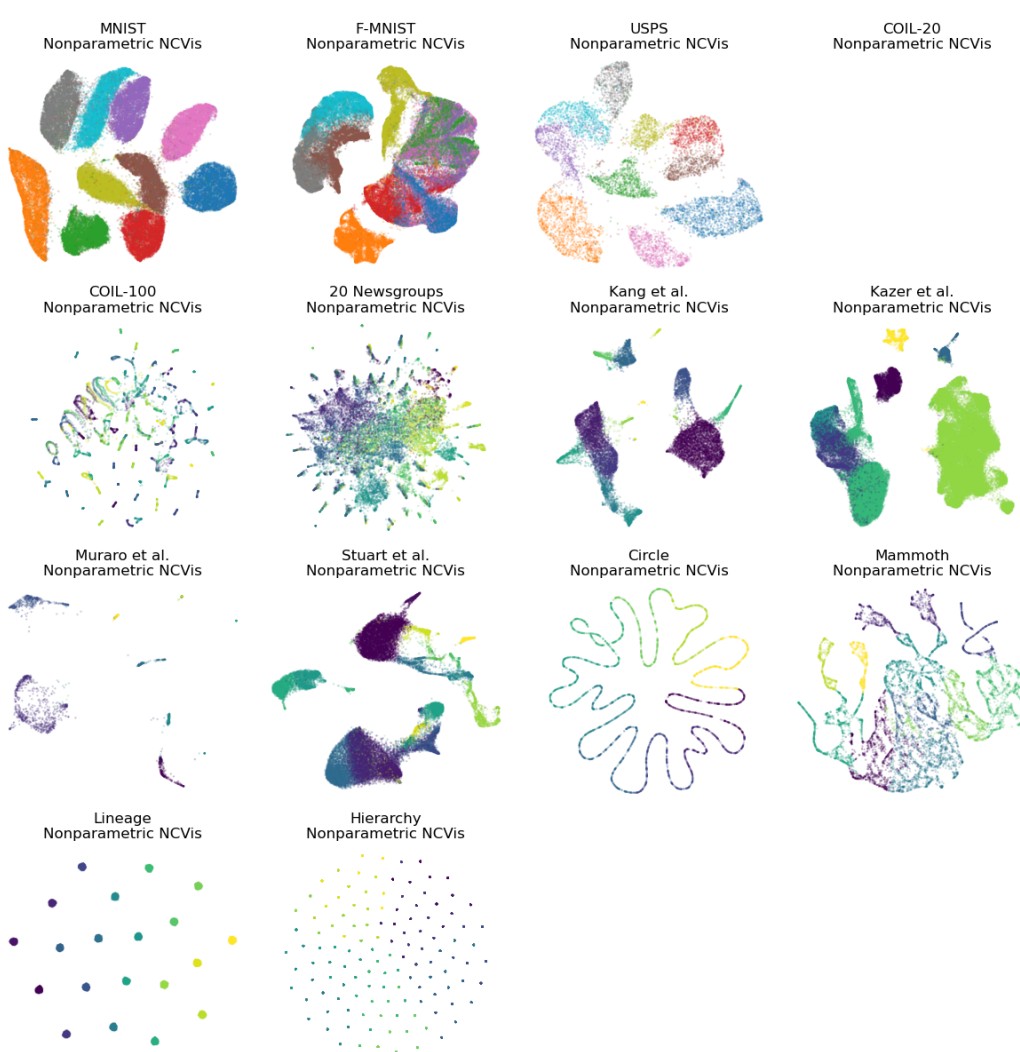

Figure 19: All dimensionality reduction results of nonparametric NCVis.

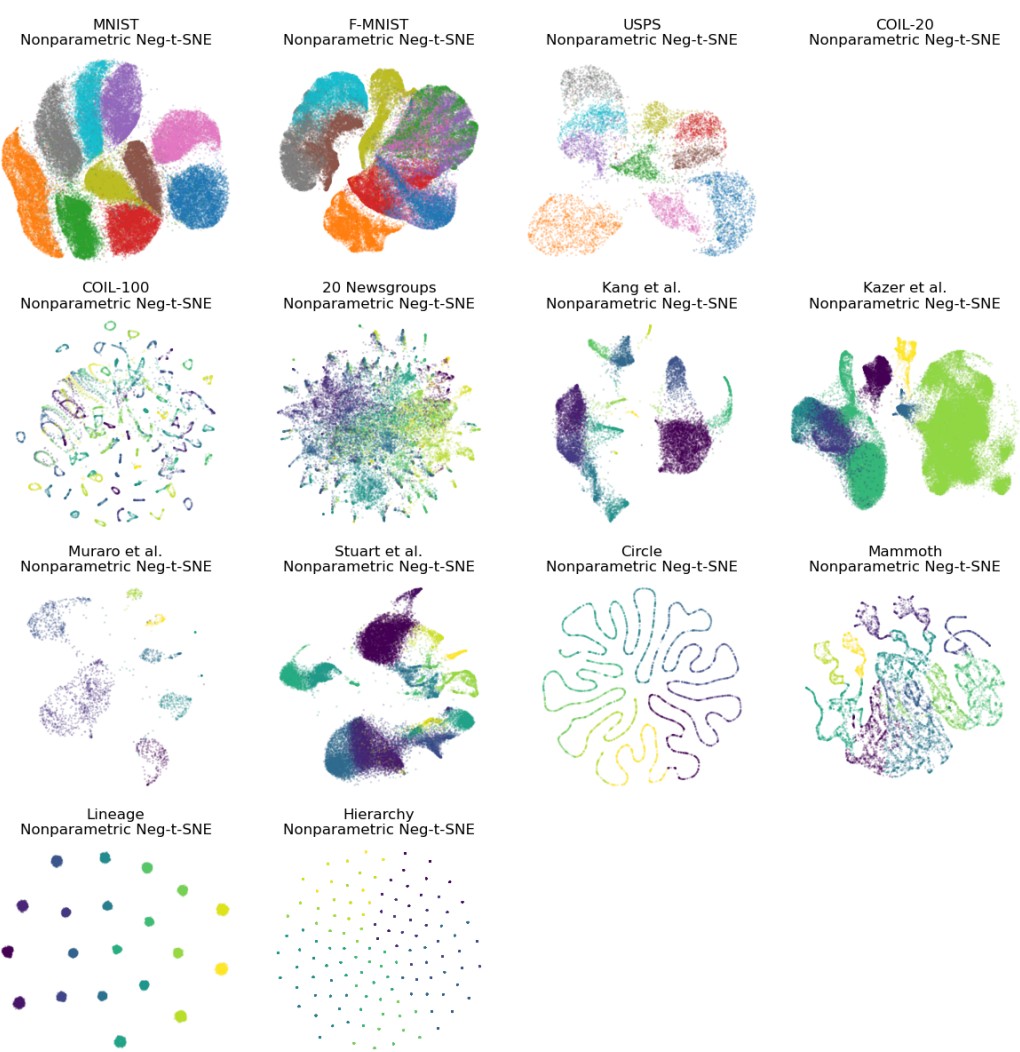

Figure 20: All dimensionality reduction results of nonparametric Neg-t-SNE.

# D Proof that PaCMAP's loss follows NEG

Recall that the NEG loss follows the form

$$\mathcal{L}^{NEG}(\theta) = -\mathbb{E}_{ij \in NN} \log \left( \frac{q_\theta(\mathbf{y}_i, \mathbf{y}_j)}{1 + q_\theta(\mathbf{y}_i, \mathbf{y}_j)} \right) - m\mathbb{E}_{ij \in \mathcal{E}} \log \left( \frac{1}{1 + q_\theta(\mathbf{y}_i, \mathbf{y}_j)} \right) \quad (10)$$

where $m$ is the number of negative samples of each batch and $q_\theta$ is the similarity function defined in the low dimensional space.

Now, we consider the PaCMAP loss. While the PaCMAP loss optimization process involves three stages with different emphasis on the loss terms, the first two stages are essentially equivalent to the early exaggeration used in t-SNE and UMAP [13]. Therefore we consider only the last stage of the PaCMAP optimization process that involves only the NN and FP losses:

$$L^{PaCMAP}(\theta) = \mathcal{L}_{ij \in NN} + \mathcal{L}_{ij \in FP} = \sum_{ij \in NN} \frac{d_2(i, j)}{d_2(i, j) + 10} + \sum_{ij \in FP} \frac{1}{d_2(i, j) + 1}. \quad (11)$$

While PaCMAP samples the repulsion using a pre-defined set of further pairs, the further pairs themselves are uniformly sampled from all the points that are not nearest neighbors. The number of neighbors are usually tiny compared to the size of the dataset. In our experiments, adding the nearest neighbors back to the further pairs candidate set does not generate any major impact to the datasets. Therefore, here we consider it to be essentially the same as sampling from $ij \in \mathcal{E}$.

Since the purpose of the loss function is to find $\theta$ that minimizes it, applying any affine transformation will not affect the optimum. Recall that in PaCMAP, the set $FP$ is $m$ times of the size of $NN$. Thus, we have:

$$L^{PaCMAP}(\theta) = \sum_{ij \in NN} \frac{d_2(i, j)}{d_2(i, j) + 10} + \sum_{ij \in FP} \frac{1}{d_2(i, j) + 1} \quad (12)$$

$$= \#NN \cdot \left( \mathbb{E}_{ij \in NN} \frac{d_2(i, j)}{d_2(i, j) + 10} + m\mathbb{E}_{ij \in \mathcal{E}} \frac{1}{d_2(i, j) + 1} \right) \quad (13)$$

$$\propto \mathbb{E}_{ij \in NN} \frac{d_2(i, j)}{d_2(i, j) + 10} + m\mathbb{E}_{ij \in \mathcal{E}} \frac{1}{d_2(i, j) + 1} \quad (14)$$

$$\propto -\mathbb{E}_{ij \in NN} \frac{10}{d_2(i, j) + 10} - m\mathbb{E}_{ij \in \mathcal{E}} \frac{d_2(i, j)}{d_2(i, j) + 1} \quad (15)$$

$$= -\mathbb{E}_{ij \in NN} \log \exp \left( \frac{10}{d_2(i, j) + 10} \right) - m\mathbb{E}_{ij \in \mathcal{E}} \log \exp \left( \frac{d_2(i, j)}{d_2(i, j) + 1} \right). \quad (16)$$

Due to the different choices of normalizing constant in the NN loss and the FP loss, the PaCMAP actually utilizes a different kernel to model the similarity between the NN and FPs. Solving for the functions $q_\theta^{NN}$ and $q_\theta^{FP}$, we have the result:

$$q_\theta^{NN}(\mathbf{y}_i, \mathbf{y}_j) = \frac{\exp(\frac{-10}{d_2(i,j)+10})}{1 - \exp(\frac{-10}{d_2(i,j)+10})} \quad (17)$$

$$q_\theta^{FP}(\mathbf{y}_i, \mathbf{y}_j) = \frac{1 - \exp(\frac{d_2(i,j)}{d_2(i,j)+1})}{\exp(\frac{d_2(i,j)}{d_2(i,j)+1})}. \quad (18)$$

$\square$

# E Proof that the Mid-Near Hard Negative False Negative Rate converges to 0 quadratically

For simplicity, we consider a dataset of size $n + 1$, so that each point will sample from a pool of $n$ points. For each point, we consider its $k = 10$ nearest neighbors as its positive points, and the rest of the points as negative points.

Now, we consider the mid-near sample process. Recall that the mid-near point samples the second closest point from a pool of 6 points. Denote the event that a mid-near point being a false negative

as $A$. $A$ essentially means that there exist more than one point from the $k$ nearest neighbors being sampled. Therefore, we know that $\bar{A}$ means there is at most one point in the samples comes from the $k$ nearest neighbors, which essentially gives us:

$$\mathbb{P}(\bar{A}) = \underbrace{\frac{\binom{n-10}{6}}{\binom{n}{6}}}_{\text{All points sampled are not NN}} + \underbrace{10 \cdot \frac{\binom{n-10}{5}}{\binom{n}{6}}}_{\text{Only one point comes from NN}} \tag{19}$$

$$= \frac{6!(n-6)!(n-10)!}{n!6!(n-16)!} + \frac{6!(n-6)!(n-10)! \cdot 10}{n!5!(n-15)!} \tag{20}$$

$$= \frac{(n-6)!(n-10)!}{n!(n-16)!} + \frac{60}{n-15} \cdot \frac{(n-6)!(n-10)!}{n!(n-16)!} \tag{21}$$

$$= \frac{n+45}{n-15} \cdot \frac{(n-6)!(n-10)!}{n!(n-16)!} \tag{22}$$

$$= \frac{(n+45)(n-10)(n-11)(n-12)(n-13)(n-14)}{n(n-1)(n-2)(n-3)(n-4)(n-5)} \tag{23}$$

$$= \frac{n^6 - 15n^5 - 1265n^4 + O(n^3)}{n^6 - 15n^5 + 85n^4 + O(n^3)}. \tag{24}$$

It follows that $\lim_{n\to\infty} \mathbb{P}(\bar{A}) = 1$, and it converges at the rate of $O(\frac{1}{n^2})$. Because $\mathbb{P}(A) = 1 - \mathbb{P}(\bar{A})$, we know that $\mathbb{P}(A)$ also converges to 0 at the rate of $O(\frac{1}{n^2})$. $\qquad\square$

This is much faster than the uniform sampling. For uniform sampling, the false negative probability is always $\frac{10}{n}$, which is linear. This is particularly bad for DR algorithms: the number of negative samples is usually linear w.r.t. $n$. Fig. 21 illustrates the probability of false negatives sampled from a dataset. We found that as long as we have more than 1330 points in our dataset, mid-near sampling can generate less false negatives than uniform sampling. This is a particularly small number to achieve in the era of big data.

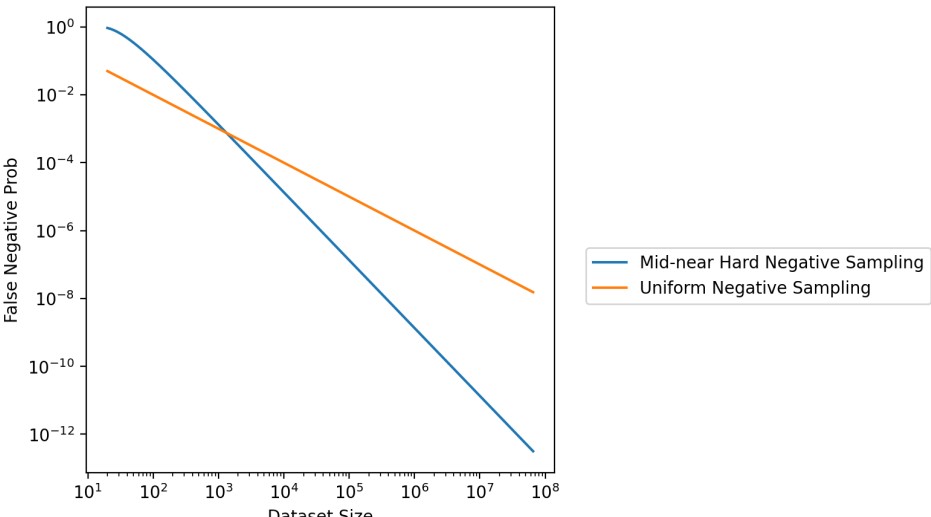

Figure 21: Expectation of the number of false negatives generated by Uniform Sampling and Mid-near Hard Negative Sampling on different dataset sizes.

## F    Implementation details for ParamRepulsor and Baseline Parametric PaCMAP

ParamRepulsor and ParamPaCMAP are implemented with PyTorch 2.0.0, Numba 0.57.0 and CUDA 11.7. We provide our implementation alongside the submission. A detailed algorithm is provided in Alg. 2.

In order to faithfully reflect the impact of parametrization on the embedding, our baseline parametric PaCMAP is written in a way to keep as much detail unchanged from the non-parametric version. ParamRepulsor implementation is written based on the Parametric PaCMAP, but many changes are made to enhance the performance on local structure.

**Network Structure.** In line with existing parametric DR method implementation [17, 18], we parametrize the projector with a shallow Multi-layer Perceptron (MLP). Unless otherwise specified, we utilize a network of three hidden layers with [100, 100, 100] neurons. ParamRepulsor utilizes SiLU as the activation function, whereas ParamPaCMAP utilizes ReLU just as the other methods. Besides utilizing basic MLP as the projector, both ParamRepulsor and ParamPaCMAP can use other network structures. We provide implementation for MLP with residual connection, convolutional neural networks, We also allow using an embedding layer as the projector so that the network behavior is similar to non-parametric version.

**Initialization.** As a non-parametric algorithm, PaCMAP directly optimizes the low-dimensional embedding, and utilizes the first two principal components of the data as its initialization. After the introduction of the neural network projector, we can no longer use this initialization. For both ParamRepulsor and ParamPaCMAP, we initialize all our neural network parameters with Kaiming Initialization [61].

**Optimization Schedule.** Since our neural network optimization schedule is performed by mini-batch stochastic gradient descent, we are unable to optimize the full embedding at once as in non-parametric PaCMAP. Therefore, at each step, we sample a batch of points, and find NN, MN, and FP points for each element in the sample.

All the points sampled will be sent to the neural network to calculate the standard PaCMAP loss. We adopt the Adam Optimizer [62] with $\beta = (0.9, 0.999)$ and a batch size of 1024. Refer to the algorithm below for more details.

# G    Experimental Details

## G.1    Datasets Used

**MNIST.** MNIST [15] is a hand-written digits dataset containing 70,000 grayscale images of the shape 28×28. The images are flattened.

**F-MNIST.** Fashion-MNIST (F-MNIST) [32] is a dataset containing 70,000 grayscale fashion images of the shape 28×28. The images are flattened.

**USPS.** USPS [63] is a dataset containing 9298 written digit images of the shape 16×16. The images are flattened.

**COIL-20** The COIL-20 [33] dataset is a is a database of 1440 gray-scale images of 20 objects. The images are flattened.

**COIL-100** The COIL-100 [34] dataset is a is a database of 7200 color images of 100 objects. The images are flattened.

**20NG** The 20NewsGroup [39] dataset contains about 18000 newsgroups posts on 20 topics. We utilize scikit-learn [64] TF-IDF vectorizer to convert each post into a vector.

**Kang et al.** The Kang et al. [35] dataset contains scRNA-seq data from 13999 cells and 14053 genes, with 13 types identified by scientists. The first 50 principal components from the raw data are used.

**Kazer et al.** The Kazer et al. [36] dataset contains scRNA-seq data from 59286 cells and 16980 genes, with 7 types identified by scientists. The first 50 principal components from the raw data are used.

---
**Algorithm 2** Detailed Pseudocode for ParamRepulsor
___
**Require:**
    $\mathbf{X}$ - high-dimensional data matrix of the shape $(N, D)$.
    $n_{NB}, n_{MN}, n_{FP}$ - the number of neighbor pairs, mid-near pairs, further pairs
    $n_{\text{epochs}}$ - the number of epochs for optimization
    $p_\theta$ - neural network projector with parameter $\theta$.
    $\eta$ - learning rate.
    $b$ - mini batch size.
    $w_{NB}, w_{MN}, w_{FP}$ – the weights associated with neighbor, mid-near, and further pairs at epoch $t$.
1:  **Initialize** neural network projector $p_\theta$ with parameter $\theta$
2:  **for** $i \leftarrow 1$ **to** $N$ **do**
3:     Sample $n_{NB}$-nearest neighbors
4:     **for** $j \leftarrow 1$ **to** $n_{MN}$ **do**
5:         Sample 6 points
6:         Select the second closest point as the $j$-th mid-near point
7:     **end for**
8:  **end for**
9:  **for** $i \leftarrow 1$ **to** $n_{\text{epochs}}$ **do**
10:     **for** $j \leftarrow 1$ **to** $n_{batches}$ **do**
11:         Sample $x = x_1 \ldots, x_b$ from training data.
12:         $x^{NN} = NN(x_1 \ldots, x_b)$ from the nearest neighbors of $x$.
13:         $x^{MN} = MN(x_1 \ldots, x_b)$ from the mid nears of $x$.
14:         $x^{FP} = FP(x_1 \ldots, x_b)$, in which $x_k^{FP} = x_t, t \sim \text{Uniform}(1, n)$.
15:         Calculate $y = f_\theta(x), y^{NN} = f_\theta(x^{NN}), y^{MN} = f_\theta(x^{MN}), y^{FP} = f_\theta(x^{FP})$.
16:         $\mathcal{L} = 0$.
17:         **for** $k \leftarrow 1$ **to** $b$ **do**
18:             $\mathcal{L} = \mathcal{L} + w_{NB} \sum_{t=1\ldots n_{NB}} \frac{d_2(y_i, y_i^{NN_t})}{10 + d_2(y_i, y_i^{NN_t})} - w_{MN} \sum_{t=1\ldots n_{MN}} \frac{d_2(y_i, y_i^{MN_t})}{1 + d_2(y_i, y_i^{MN_t})} -$
            $w_{FP} \sum_{t=1\ldots n_{FP}} \frac{d_2(y_i, y_i^{FP_t})}{1 + d_2(y_i, y_i^{FP_t})}$.
19:         **end for**
20:         Calculate gradients $\nabla_\theta \mathcal{L}$.
21:         Update parameters $\theta$ using Adam optimizer.
22:     **end for**
23:  **end for**
24:  **return** $f_\theta(\mathbf{X})$
___

**Muraro et al.**    The Muraro et al. [37] dataset contains scRNA-seq data from 2282 cells and 18962 genes, with 9 types identified by scientists. The first 50 principal components from the raw data are used.

**Stuart et al.**    The Stuart et al. [38] dataset contains scRNA-seq data from 30672 cells and 17009 genes, with 25 types identified by scientists. The first 50 principal components from the raw data are used.

**Circle**    The Circle dataset comprises of 5000 points uniformly sampled from a 2D circle with radius 1. The circle is divided into ten arcs of the same length, and each point receives a label that represents the index of the arc it belongs to.

**Mammoth**    The mammoth dataset [42, 41] contains 10k points from a 3D woolly mammoth skeleton.

**Lineage**    The Gaussian Lineage dataset [5, 43] contains 10000 points in twenty 50-dimensional Gaussians, equally separated on a line.

**Hierarchy**    The Gaussian Hierarchical Dataset [43] contains 12500 points. The points belongs to 125 micro clusters, arranged into 5 macro and 25 meso clusters. Each micro cluster includes 100 observations.

### G.2 Computation Platforms

All experiments are conducted with an Exxact TensorEX 2U Server with 2 Intel Xeon Ice Lake Gold 5317 Processors @ 3.0GHz. We limit the RAM usage to be 32GB. Parallel computation are performed over a single Nvidia RTX A5000 GPU.

## H  Additional Experiments, Tables and Figures

### H.1  Additional Analysis on Distance Distribution in Embedding

Fig. 22 provides a comprehensive analysis over distances between different kinds of pairs, generated by multiple DR algorithms. We can see that all parametric methods generate a shorter FP distance compared against their non-parametric counterpart. MN pairs, though should be classified as FPs, tend to be harder to optimize, resulting in a shorter distance on average.

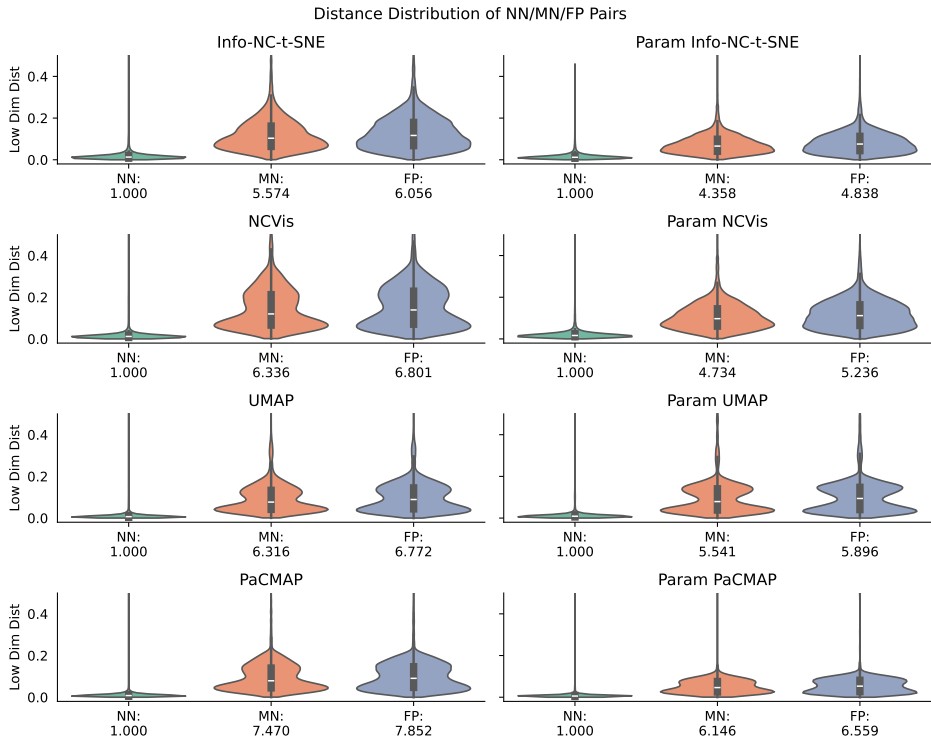

Figure 22: The low-dimensional scaled distance distribution between various types of point pairs with labels "3" and "8" in the embedding of the MNIST digit dataset [15], generated by Info-NC-t-SNE, NCVis, UMAP, PaCMAP, and their parametric counterpart. See definition in Sec. 2 & 4.

### H.2  Computational Speed Evaluation

As datasets grows larger, scalability becomes more important. We evaluate the time consumed by ParamInfo-NC-t-SNE, ParamRepulsor and ParamUMAP on two extremely large datasets, from [65] and [4]. The dataset sizes are $1,306,127$ and $2,058,652$, respectively. The results are shown in Figure 23. We can see that ParamRepulsor outperforms ParamUMAP in terms of scalability. While ParamRepulsor is slower than ParamInfo-NC-t-SNE, the speed is still comparable in terms of magnitude. We note that ParamInfo-NC-t-SNE utilizes smaller number of epochs, which gives it a higher speed, but at the cost of an underoptimized embedding, as shown in Section 5. The computational efficiency of ParamRepulsor can be further improved by a better optimization schedule as well as computational improvements, which we leave for future works.

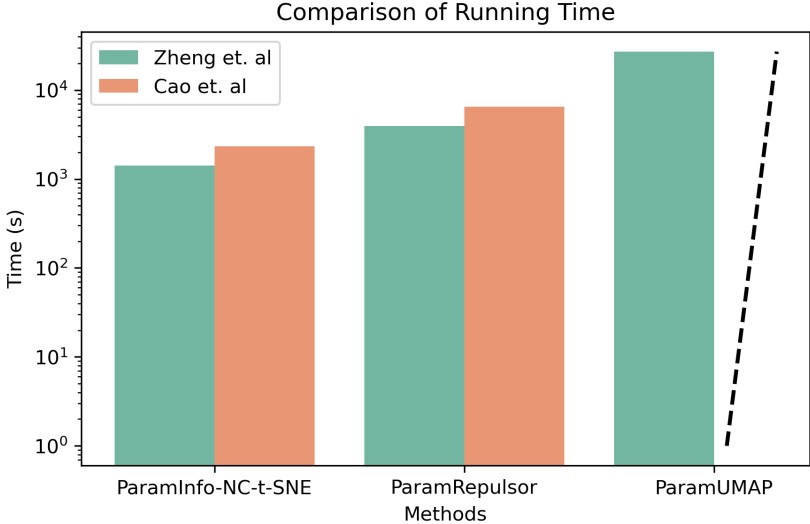

Figure 23: Time consumed by parametric DR methods compared to the size of the dataset. ParamUMAP cannot finish the Cao et. al dataset under the time constraint of 6 hours.

## H.3  Additional Tables

Table 2, 3, 5, 4 measure the SVM accuracy, $k$-nearest neighbor preservation ratio, Triplet preservation ratio, and cluster centroid distance correlation. We note that GeoAE performs particularly well on Triplet preservation. This is expected: as an autoencoder-based method, GeoAE aims to preserve the geographical distance information in the high-dimensional space, usually at the cost of the local structure. As a result, its local structure performance is particularly low. However, our method, ParamPaCMAP and ParamRepulsor, achieve comparable result on this metric.

Table 2: SVM Accuracy of DR methods measured on various datasets. The absence of values indicates that the method failed to produce a valid embedding.

| METHOD | P-UMAP | P-ItSNE | P-NtSNE | P-NCVis | GeoAE | P-PaCMAP | P-Rep |
|---|---|---|---|---|---|---|---|
| MNIST | *0.964* | 0.836 | 0.865 | 0.836 | 0.787 | *0.966* | **0.968** |
| F-MNIST | *0.725* | *0.716* | *0.716* | 0.640 | 0.714 | *0.719* | **0.749** |
| USPS | *0.953* | 0.931 | *0.934* | 0.933 | 0.835 | *0.948* | **0.955** |
| COIL-20 | *0.813* | – | – | – | 0.661 | *0.822* | **0.856** |
| COIL-100 | 0.237 | – | – | – | 0.493 | 0.825 | **0.862** |
| 20NG | **0.462** | 0.355 | 0.384 | *0.416* | 0.065 | *0.419* | *0.457* |
| KANG | *0.931* | *0.936* | *0.936* | *0.932* | 0.482 | *0.947* | **0.955** |
| KAZER | **0.938** | *0.935* | *0.935* | *0.936* | 0.758 | *0.930* | *0.935* |
| MURARO | *0.955* | **0.961** | **0.961** | *0.957* | 0.589 | *0.960* | **0.961** |
| STUART | 0.768 | *0.832* | **0.834** | *0.832* | 0.425 | 0.789 | *0.832* |
| CIRCLE | *0.899* | *0.904* | *0.902* | **0.910** | *0.894* | *0.905* | *0.894* |
| MAMMOTH | *0.902* | 0.886 | 0.886 | *0.891* | **0.936** | 0.887 | 0.895 |
| LINEAGE | **1.000** | *0.999* | *0.999* | **1.000** | **1.000** | **1.000** | **1.000** |
| HIERARCHY | 0.345 | 0.424 | 0.424 | 0.357 | 0.200 | 0.555 | **0.622** |

**Additional Global Structure Preservation: Preservation of Triplets** Preservation of global structure also involves the preservation of distances. Particularly, we would like the relative relationship of distances to be preserved: if point A and B are closer than A and C in the high dimensional space, they should be closer in the embedding as well. Following [13], we evaluate each method's ability to preserve distance relationships between randomly sampled triplets. The result is in Table 5 in App. H.3.

Table 3: Ratio of 30-NN kept by the embedding measured on various datasets.

| METHOD | P-UMAP | P-ITSNE | P-NTSNE | P-NCVIS | GEOAE | P-PACMAP | P-REP |
|---|---|---|---|---|---|---|---|
| MNIST | *0.084* | 0.055 | 0.071 | 0.038 | 0.074 | *0.090* | **0.106** |
| F-MNIST | *0.088* | 0.079 | 0.044 | 0.043 | *0.081* | *0.097* | **0.121** |
| USPS | **0.317** | *0.313* | 0.286 | 0.298 | 0.224 | *0.280* | *0.306* |
| COIL-20 | *0.710* | – | – | – | 0.490 | *0.701* | **0.713** |
| COIL-100 | 0.067 | – | – | – | 0.348 | *0.523* | **0.593** |
| 20NG | **0.220** | 0.156 | *0.180* | *0.192* | 0.003 | 0.140 | 0.105 |
| KANG | *0.100* | *0.113* | *0.100* | *0.111* | 0.008 | *0.094* | **0.121** |
| KAZER | 0.051 | *0.059* | 0.050 | *0.057* | 0.004 | 0.047 | **0.065** |
| MURARO | *0.393* | *0.416* | 0.368 | 0.321 | 0.062 | *0.387* | **0.429** |
| STUART | 0.081 | **0.099** | *0.086* | *0.098* | 0.005 | *0.083* | **0.099** |
| CIRCLE | **0.901** | *0.896* | *0.899* | *0.896* | *0.895* | *0.898* | **0.901** |
| MAMMOTH | *0.559* | *0.559* | *0.552* | *0.555* | **0.593** | 0.545 | *0.571* |
| LINEAGE | *0.077* | **0.095** | *0.076* | *0.094* | *0.076* | *0.076* | **0.095** |
| HIERARCHY | *0.367* | **0.370** | *0.364* | *0.367* | 0.352 | *0.362* | *0.365* |

Table 4: Centroid Distance Correlation of DR methods measured on various datasets. The absence of values indicates that the method failed to produce a valid embedding. The highest values are displayed in bold. Values that has no significant difference from the highest (measured by an independent t-test) are shown in italics.

| METHOD | P-UMAP | P-ITSNE | P-NTSNE | P-NCVIS | GEOAE | P-PACMAP | P-REP |
|---|---|---|---|---|---|---|---|
| MNIST | 0.707 | 0.697 | 0.705 | 0.663 | *0.759* | **0.784** | *0.732* |
| F-MNIST | 0.920 | *0.897* | *0.897* | 0.894 | 0.864 | **0.922** | *0.907* |
| USPS | *0.911* | 0.851 | *0.910* | **0.917** | 0.843 | *0.879* | 0.816 |
| COIL-20 | *0.538* | – | – | – | 0.780 | *0.767* | **0.824** |
| COIL-100 | 0.607 | – | – | – | 0.677 | 0.697 | **0.760** |
| 20NG | 0.751 | *0.811* | 0.799 | 0.768 | 0.319 | **0.832** | 0.686 |
| KANG | 0.549 | **0.556** | **0.556** | 0.515 | *0.547* | *0.521* | 0.457 |
| KAZER | **0.682** | 0.647 | 0.643 | *0.678* | 0.554 | 0.618 | 0.544 |
| MURARO | 0.754 | **0.795** | 0.776 | 0.755 | 0.587 | 0.658 | 0.576 |
| STUART | 0.292 | 0.391 | 0.329 | 0.357 | 0.295 | **0.432** | 0.322 |
| CIRCLE | *0.957* | 0.879 | *0.898* | **0.969** | *0.937* | *0.918* | *0.953* |
| MAMMOTH | *0.929* | 0.986 | **0.987** | *0.974* | 0.908 | *0.972* | *0.972* |
| LINEAGE | **1.000** | **1.000** | **1.000** | **1.000** | *0.996* | **1.000** | **1.000** |
| HIERARCHY | 0.350 | 0.464 | 0.571 | *0.489* | 0.647 | *0.708* | **0.738** |

Table 5: Triplet Preservation of DR methods measured on various datasets. The absence of values indicates that the method failed to produce a valid embedding. The highest values are displayed in bold. Values with no statistically significant difference from the highest (as determined by an independent t-test) are highlighted in italics.

| METHOD | P-UMAP | P-ITSNE | P-NTSNE | P-NCVIS | GEOAE | P-PACMAP | P-REP |
|---|---|---|---|---|---|---|---|
| MNIST | *0.600* | *0.611* | *0.615* | 0.588 | **0.628** | *0.604* | *0.605* |
| F-MNIST | *0.720* | *0.738* | *0.738* | *0.747* | **0.789** | *0.722* | 0.706 |
| USPS | *0.663* | *0.665* | *0.669* | *0.672* | **0.686** | *0.651* | *0.658* |
| COIL-20 | 0.612 | – | – | – | **0.739** | *0.678* | *0.719* |
| COIL-100 | 0.615 | – | – | – | **0.730** | *0.687* | *0.720* |
| 20NG | *0.655* | *0.674* | *0.666* | *0.658* | 0.528 | **0.678** | 0.607 |
| KANG | *0.772* | 0.746 | *0.755* | *0.775* | 0.638 | **0.792** | *0.772* |
| KAZER | *0.768* | *0.770* | *0.771* | *0.774* | 0.752 | **0.784** | *0.761* |
| MURARO | 0.693 | 0.719 | *0.717* | *0.721* | 0.663 | **0.763** | 0.742 |
| STUART | 0.628 | 0.688 | 0.689 | 0.660 | 0.629 | **0.739** | 0.713 |
| CIRCLE | *0.980* | 0.900 | 0.905 | **0.983** | 0.945 | 0.932 | *0.975* |
| MAMMOTH | 0.878 | *0.933* | **0.934** | *0.915* | 0.864 | *0.917* | 0.905 |
| LINEAGE | **0.995** | **0.995** | **0.995** | **0.995** | *0.991* | *0.994* | *0.993* |
| HIERARCHY | 0.625 | 0.707 | *0.736* | 0.674 | *0.741* | *0.788* | **0.789** |

