# OpenReview forum: "Navigating the Effect of Parametrization for Dimensionality Reduction"
_NeurIPS.cc/2024/Conference — NeurIPS 2024 poster_

### Official Review · Reviewer_QK23 · 2024-07-08

**Soundness:** 3
**Presentation:** 3
**Contribution:** 2
**Rating:** 6
**Confidence:** 3

**Summary:**

The paper focuses on comparing parametric and non-parametric neighborhood embedding methods for dimensionality reduction. Non-parametric methods excel at preserving local data structures but struggle with large datasets. Parametric methods, utilizing neural networks, offer better scalability but often compromise local structure preservation. The paper reveals that parametric methods tend to prioritize global structure preservation at the expense of local details, leading to less distinct cluster boundaries. This is attributed to their inability to effectively repulse negative pairs and sensitivity to the choice of loss function.

To address these issues, the authors propose a new parametric method called ParamRepulsor with two key improvements:

Hard Negative Mining: This technique focuses on identifying and repulsing the most challenging negative pairs, improving the separation between dissimilar points.
Modified Loss Function: The authors use a loss function that applies a stronger repulsive force, particularly on hard negative pairs, leading to better local structure preservation.

In experiments, ParamRepulsor is shown to outperform other parametric methods on various datasets.

**Strengths:**

Novel Insight: The paper provides a novel perspective on the differences between parametric and non-parametric dimensionality reduction methods, highlighting the issue of local structure preservation in parametric methods.

Technical Contribution: The introduction of ParamRepulsor, a new parametric method with hard negative mining and a modified loss function, is a significant technical contribution. The paper demonstrates its superior performance in preserving local structures compared to other parametric methods.

Thorough Evaluation: The paper includes a comprehensive evaluation of ParamRepulsor and other dimensionality reduction methods on a diverse range of datasets, providing strong empirical evidence for the effectiveness of the proposed approach.

Clear Presentation: The paper is well-written and easy to read.

**Weaknesses:**

Limited Generalizability: The focus of this paper is on parametric DR methods that utilize shallow neural networks. However, the generalizability of the findings to other parametric models, such as deep neural networks, remains unexplored. This is particularly relevant as the paper's results indicate that all parametric methods tend to better preserve local structures, and the performance differences between algorithms diminish as network depth increases.

Insufficient Theoretical Analysis: While the paper offers empirical evidence, it lacks a comprehensive theoretical analysis. It remains unclear why the same algorithm that performs well in its non-parametric version underperforms in the parametric version (loses local structure). Although the authors acknowledge that the limited capacity of shallow neural networks is widely accepted as an explanation for their failure, they dismiss it as the primary cause.

Absence of Non-Parametric Repulsor: The paper presents both parametric and non-parametric versions for all algorithms except ParamRepulsor. Including a non-parametric version of ParamRepulsor would provide better intuition into how the modified terms (e.g., including MN terms) affect the results. Additionally, the paper does not theoretically analyze why hard negative mining and the specific loss function are effective.

Limited Practical Impact/Advantage: While the paper demonstrates the superior performance of ParamRepulsor compared to other parametric methods in the shallow network case, this advantage might not be significant when deeper networks are used. This limitation could potentially reduce the practical impact and contribution of this work.

Evaluation Metrics: The evaluation metrics used in the paper, although standard in the field, may not fully capture all aspects of dimensionality reduction quality, especially for complex datasets.

**Questions:**

Network Depth and Embedding Structure: The paper exclusively focuses on shallow neural networks for parametric dimensionality reduction. It would be valuable to understand the rationale behind this choice. Is there a risk of overfitting with deeper networks? The authors mention that "The shallow parametrization used in NE DR methods ensures that distances in the high-dimensional space remain correlated with those in the low-dimensional embedding." Investigating how the embedding structure changes with increasing network depth could provide valuable context for interpreting the results of this work.

**Limitations:**

While the authors transparently acknowledge certain limitations, such as the need for further theoretical analysis and computational optimization of ParamRepulsor, other significant limitations remain unaddressed.

The paper's exclusive focus on shallow networks limits the generalizability of its findings. The authors do not provide a clear rationale for this choice, nor do they explore the impact of network depth on the preservation of local structures. This is particularly important given the observation that deeper networks might naturally mitigate the issues observed with shallow parametric models.

Suggestions to enhance the impact and relevance of this work:

Expanding the scope: Investigate the behavior of ParamRepulsor and other parametric methods with deeper networks.
Developing Non-Parametric Repulsor: Create a non-parametric version of ParamRepulsor for direct comparison and to isolate the effect of the proposed modifications.
Deeper Theoretical Analysis: Conduct a more in-depth analysis to understand the underlying reasons for the performance differences between parametric and non-parametric methods and the impact of network depth on local structure preservation.

---

> ### Author Rebuttal · Authors · 2024-08-06
>
> We thank the reviewer for the very detailed and encouraging review, especially the suggestions made to enhance our paper. Here are our detailed response to each question or concern raised by the reviewer:
>
> - **Rationale behind the choice for shallow neural network, and generalizability to deep neural networks.**
>
> Increasing the number of layers empirically gives diminishing returns in terms of local quality improvement. While the performance gap between the non-parametric and parametric method reduces when we increase the number of layers from 0 to 3, such performance gaps stop to reduce when we further increase the number of layers. Fig. 2 in the common response PDF compares the performance of P-ItSNE, P-UMAP and P-PaCMAP with 4 or 5 layers. P-ItSNE receives minimal increase in quality, whereas for P-UMAP and P-PaCMAP, the embeddings do not change significantly. In all three cases, the performance is not as good as their non-parametric counterpart. Given that increasing the number of layers significantly hurts scalability, which is important for DR algorithms, only shallow neural networks were used in our main experiments. See common response Fig. 3 for discussion on other network architecture.
>
> - **Why is the limited capacity of the shallow neural network not the sole reason for the loss of local structure?**
>
> While we believe that parametrization contributes to this issue, as stated on lines 211-212, it is not the only factor. As shown in Figure 2 of the response PDF, increasing the number of layers does not completely restore the local structure. If network capacity were the sole issue, we would expect a complete restoration of local structure with additional layers.
>
> We found that the problem in DR methods is similar to issues identified in ordinary contrastive learning by [1], where the repulsive force is insufficient. This problem also resonates with the violation of the strong repulsion principle of DR discussed by [2]. Therefore, we introduced hard negative mining into DR to solve this problem, which is used in contrastive learning for the same purpose - they also don’t have a theoretical explanation for it.
>
> The closest we can get to a theoretical explanation is to refer back to the theoretical principles in [2], specifically Principle 5 states that the repulsion force needs to be strong when further pairs are close (gradient should be strong when further pairs are close), and Principle 2 states that the gradient should be weak when further pairs are far apart. This means scaling is important: if the scaling is incorrect, we have a weak repulsive force when we should have a strong one. We believe this is the problem with parametric approaches - they produce an incorrect scale. We tried several different ways to handle this, including rescaling the axes, but the hard negatives remedy the problem more effectively because they directly increase the gradients for informative negative pairs so that Principles 2 and 5 are obeyed.
>
> - **How does our optimization perform in the non-parametric setting?**
>
> Please check common response 2.
>
> - **Evaluation with more metrics.**
>
> We thank the reviewer for acknowledging that evaluation metrics used in the paper are standard in the field. Beyond the empirical results discussed in the main text, we also conducted experiments on other global-based metrics in Appendix Sec. F.  Indeed what makes DR challenging to evaluate is that there is no single metric. Many people have thought about this, and many papers have been written about this issue [3-7]. Consequently, we adhere to the current standards in the field.
>
> [1] Robinson, Joshua, et al. "Contrastive learning with hard negative samples." ICLR 2021.
>
> [2] Wang, et al. "Understanding how dimension reduction tools work: an empirical approach to deciphering t-SNE, UMAP, TriMAP, and PaCMAP for data visualization." JMLR 2021.
>
> [3] Kobak, et al. "The art of using t-SNE for single-cell transcriptomics." Nature communications 10.1 (2019): 5416.
>
> [4] Cakir, et al. "Comparison of visualization tools for single-cell RNAseq data." NAR Genomics and Bioinformatics 2.3 (2020).
>
> [5] Sun, et al. "Accuracy, robustness and scalability of dimensionality reduction methods for single-cell RNA-seq analysis." Genome biology 20 (2019): 1-21.
>
> [6] Nguyen, et al. "Ten quick tips for effective dimensionality reduction." PLoS computational biology 15.6 (2019): e1006907.
>
> [7] Espadoto, et al. "Toward a quantitative survey of dimension reduction techniques." IEEE transactions on visualization and computer graphics 27.3 (2019): 2153-2173.

---

> > ### Comment · Reviewer_QK23 · 2024-08-13
> > **Thank you for your response**
> >
> > I thank the authors for the clarification and the extra figures in their response.  I'm raising the score to 6.

---

### Official Review · Reviewer_bihR · 2024-07-11

**Soundness:** 1
**Presentation:** 3
**Contribution:** 1
**Rating:** 2
**Confidence:** 5

**Summary:**

Authors study the effect of parameterization in neighbor embedding algorithms. They aim to improve how these methods capture local structure in the data. To do so they propose a new algorithm building upon PaCMAP with slight modification of its loss with new weights in front of each term.

**Strengths:**

* code is provided.
* the experimental section is rich with a good variety of datasets and metrics. Conducted experiments seem to make sense.
* section 2 is clear and efficient.

**Weaknesses:**

I am disappointed that this work does not really dive into the effects of amortization on neighbor embeddings. I was expecting more discussions on the effect of batch size, initialization, learning rates etc... Many discussions here also seem to apply to the non parametric setting, we often have the impression to move away from the initial focus of this work. The problem at hand is not correctly introduced in section 3 which makes the paper unclear to follow and understand. Here are some issues:

* the sampling procedure to obtain mid-near pair is taken from PACMAP is not a new contribution.
* it would be helpful to state more clearly the difference between the proposed objective and the one of PACMAP. From what I understand it amounts to switching the sign of the mid-near term from attraction to repulsion. To me it not clear at all how is this change related to parameterization. It would be interesting to see how the new loss function would perform in the non-parametric setting as well.
* About observation 2 in section 3, it is stated that NEG based methods perform better in local structure preservation and : *This is in contrast to both NCE and InfoNCE losses, where each negative pair term is based on all pairs.* . However, reading through : https://arxiv.org/pdf/2206.01816 , I have the understanding than even NEG based algorithms sample negatives pairs uniformly in all pairs. Therefore this point is not clear to me. Also I am wondering why it is related to the parametric setting and not non-parametric as well.
* In the experimental part it would also be interesting to compare with non parametric methods which are expected to perform better. This way we could measure how does this paper contribute to closing the gap between parametric and non parametric.
* one major drawback of this approach is that there are too many hyperparameters to tune with all the weight terms in the loss. Looking at the code we have : `def paramrepulsor_weight_schedule(epoch):
    if epoch < 200:
        w_nn = 4.0
        w_fp = 8.0
        w_mn = 0.0
    else:
        w_nn = 1.0
        w_fp = 8.0
        w_mn = -12.0
    weight = np.array([w_nn, w_fp, w_mn])
    return weight`. How were these values determined ? do they need to be tuned for each dataset ? From the practitioner point of view, I have strong doubts about the practical utility of such machinery.

More minor comments:
* false negatives should be clearly defined before theorem 4.1 .
* theorem 4.1 is an asymptotic result thus it doesn't bring true insights about practical guarantees of the methods.
* I wouldn't put Corollary 4.2. in the form of a corollary as there is no associated proof.

**Questions:**

* in the observation 1 section 3 :  *all the parametric counterparts (App. F.1) have a smaller FP distance ratio, meaning further pairs are positioned closer together* . Woul you have more insights on why this phenomenon happen ? how robust is this observation to batch size, learning rate and architectures used ?
* Do observation 3 and 4 also apply in the non-parametric setting ? Could you discuss more the effect of parametrization on these.
* in algorithm 1 we can see that the kNN graphs are constructed once at the beginning. Would there be a way to adapt this construction to the online setting *i.e.* to easily update these graphs during iterations?
* did the authors consider using entropic affinity graphs as in TSNE instead of binary kNN graphs ? these affinities can be seen as smooth kNN graphs and they are usually faster to compute that kNN on GPUs as they are highly parallelizable.

**Limitations:**

/

---

> ### Author Rebuttal · Authors · 2024-08-06
>
> We thank the reviewer for the detailed review. Here are our response to each question or concern:
>
> - **Mid-near sampling is not new.**
>
> While the sampling process of the hard negatives roots from the mid-near sampling proposed by PaCMAP, in our work, we utilize them for a completely different purpose - and MN pairs even create forces in the opposite direction in our work. PaCMAP utilizes the mid-near pairs to increase global structure preservation, whereas in our work, we use it to encourage separation of clusters particularly in the parametric setting.
>
> - **Unclear how the proposed change affects the parametrization. Will this change behave similarly in non-parametric settings as well?**
>
> See common response 2.
>
> - **What does it mean for NEG loss to have a negative term that is not based on all pairs? How would it affect the non-parametric case?**
>
> To clarify, we are discussing the calculation of the loss for each individual pair, not the sampling process. For NEG loss, the loss and gradient for each individual pair are independent of the distances of any other pair. This independence is useful in the parametric case because the neural network projector correlates high-dimensional and low-dimensional distances, leading to potential inaccuracies in estimating the average distance between repulsive pairs. In the case of NCE loss, the loss value of any individual pair is determined by the pair's distance and a global average. This dependency prevents the generation of sufficient repulsive gradients.
>
> - **Comparison against non-parametric methods.**
>
> See common response 1.
>
> - **Too many hyperparameters to tune.**
>
> To clarify, the piece of code you mentioned is actually an optimization parameter schedule that is **not** meant to be tuned by users on each dataset. Applying a multi-stage loss schedule during the optimization process is a common practice in ML and DR. Examples include the most prominent algorithms in this field, including t-SNE [1], UMAP [2], and recent algorithms such as FIt-SNE [3], PaCMAP [4] and InfoNC-t-SNE [5]. All of these methods have numerous parameters to set, it’s just that the user often doesn’t have the option to control them. Sections 3 and 4 explain why our parameters are set as they are: the NN loss is decreased similarly to t-SNE's early exaggeration, and the hard negative loss is increased to overcome the lack of gradient in repulsion pairs due to parametrization. Empirical evaluation across 14 datasets coming from four different modalities show that our optimization schedule (all parameters unchanged across all datasets) can consistently produce embedding of high quality agnostic to the data. The results are robust to small to moderate changes in the parameter values, indicating that the general forces and balance between terms are what truly matter.
>
> - **False negative definition and validity of Theorem 4.1.**
>
> In this context, a false negative is defined as a pair of points that should be nearest neighbors but are incorrectly sampled as a negative pair. To illustrate the effectiveness of theorem 4.1, we conduct a new empirical experiment with 10 repeated trials on the MNIST dataset. The results show that, when sampling 20 pairs for each point, a naive sampling method results in an average of 201.5 false negative pairs. In contrast, our hard negative sampling approach reduces this to an average of 0.8 false negatives.
>
> - **How robust is observation 1 under different network settings?**
>
> Observation 1 is pretty robust across different learning rates, batch sizes, and architecture in our experiments. Due to space constraints, Figure 3 in the shared PDF only visualizes ParamInfo-NC-t-SNE with different learning rates (3e-4, 1e-3, 3e-3), ParamPaCMAP with different network architectures (3-layer MLP with 100 neurons, 3-layer MLP with residual connections, a tiny 3-layer CNN), and ParamUMAP with different batch sizes (1024, 2048, 4096). All nine cases demonstrated worse visual effects and local structure preservation, verifying our observation.
>
> - **Do observations 3 and 4 apply to non-parametric cases?**
>
> Yes, observation 3 and 4 still holds for non-parametric cases, as the pair sampling is separate from whether the model is parametric or non-parametric. It should be noticed that non-parametric algorithms do not suffer from the gradient diminishing problem, so applying the Repulsor changes are unnecessary for non-parametric cases.
>
> - **Is it possible to extend the k-NN graph construction to the online setting?**
>
> It is possible to update the embedding, but certain difficulties remain. The graph needs to be reconstructed, and newly added points may become k-NN of points in the original data. Consequently, the embeddings of the original points no longer represent the similarities between neighbors and need to be updated as well. In contrast, parametric methods do not experience such problems.
>
> - **Applying entropy affinity graphs as in t-SNE?**
>
> We choose to use k-NN graphs out of concerns on scalability, since naive affinity computation for all pairs in a dataset of size N will lead to a space complexity of $O(N^2)$, and a computational complexity of $O(N^2logN)$. Nevertheless, we note that recent methods such as SNEKhorn [6] have successfully extended the affinity graph approach with GPU. We will include discussion on these works into the related work section if the paper is accepted.
>
> [1] Van der Maaten, et al. "Visualizing data using t-SNE." JMLR (2008).
> [2] McInnes, et al. "Umap: Uniform manifold approximation and projection for dimension reduction." arXiv:1802.03426.
> [3] Linderman, et al. "Fast interpolation-based t-SNE for improved visualization of single-cell RNA-seq data." Nature Methods (2019)
> [4] Wang, et al. "Understanding how dimension reduction tools work." JMLR 2021
> [5] Damrich, et al, “From t-SNE to UMAP with contrastive learning”, ICLR 2023
> [6] Van Assel, et al. "SNEKhorn: Dimension reduction with symmetric entropic affinities." NeurIPS 2023.

---

> > ### Comment · Reviewer_bihR · 2024-08-12
> >
> > Thank you for the rebuttal.
> >
> > I still have several key concerns regarding this work:
> >
> > **The proposed method is limited to PACMAP.**
> >
> > The proposed ParamRepulsor and ParamPaCMAP are minor adaptations of the PACMAP algorithm, which is not among the most widely used neighbor embedding methods. In my view, focusing on enhancing the parametric versions of more popular algorithms like tSNE, UMAP, or LargeVis would have a much greater impact. As a result, the scope of this paper feels very limited.
> >
> > **The motivations for parametric neighbor embedding are unclear.**
> >
> > Neighbor embedding algorithms are incredibly efficient. Embedding a few million points can be done in a matter of seconds on a GPU using recent implementations of TSNE or LargeVis. However, this efficiency is still far from what we see in other domains like SSL, where scalability and efficiency present interesting research directions. This leads me to question the importance of developing parametric methods for neighbor embedding, particularly when practitioners typically recompute the full embeddings when new data is introduced.
> >
> > I'm curious if the authors can provide references where an amortized, differentiable model for neighbor embedding is crucial. To my knowledge, these methods are primarily used for data exploration rather than being integrated into larger pipelines that would benefit from differentiability, as is common in SSL.
> >
> > This point further limits the scope of the paper.
> >
> > **Lack of theoretical support for the method.**
> >
> > In my view, this method adds unnecessary complexity and introduces numerous steps without sufficient justification. The arguments seem to rely more on intuition than on solid theoretical backing.
> >
> > As a practitioner, I already find PACMAP challenging due to its numerous parameters, and this approach seems to further complicate the field unnecessarily. I strongly disagree with the authors' assertion that other methods have similar parameter complexity.
> >
> > From my experience, the methods that gain practical adoption are those that are straightforward to implement and easy for practitioners to grasp, minimizing the use of tricks and adhering to core principles. This is especially true in unsupervised learning, where evaluating model performance is difficult. While simplicity might come at the cost of slight performance trade-offs on the toy datasets often used in ML research, principled methods tend to perform better on real-world datasets encountered by practitioners.
> >
> > TSNE is a prime example, consistently delivering state-of-the-art results across a wide range of datasets. Similarly, LargeVis, which extends TSNE with negative sampling, is another method that is accessible to ML practitioners.
> >
> > The paper in question takes the opposite approach, which is why I believe it does not contribute value to the field of dimensionality reduction.
> >
> > For these reasons, I am convinced that this work does not meet the bar for NeurIPS, and I am modifying my score accordingly.

---

> > > ### Author Response · Authors · 2024-08-12
> > >
> > > > The motivations for parametric neighbor embedding are unclear.
> > >
> > > > Neighbor embedding algorithms are incredibly efficient. Embedding a few million points can be done in a matter of seconds on a GPU using recent implementations of TSNE or LargeVis. However, this efficiency is still far from what we see in other domains like SSL, where scalability and efficiency present interesting research directions. This leads me to question the importance of developing parametric methods for neighbor embedding, particularly when practitioners typically recompute the full embeddings when new data is introduced.
> > >
> > > > I'm curious if the authors can provide references where an amortized, differentiable model for neighbor embedding is crucial. To my knowledge, these methods are primarily used for data exploration rather than being integrated into larger pipelines that would benefit from differentiability, as is common in SSL.
> > >
> > > > This point further limits the scope of the paper.
> > >
> > >
> > > While we agree that nonparametric methods are indeed efficient, we believe that **efficiency is not the only criterion** to consider. Both nonparametric and parametric methods serve important roles in data exploration and analysis. In terms of uses, both nonparametric methods and parametric methods can be used for data exploration as well as analysis. However, parametric methods offer unique advantages, particularly when a map $x\rightarrow y$ between input and output spaces is needed. This capability is crucial for **comparing multiple datasets or adding new data to an existing embedding**, as demonstrated in our experiments. They can also handle global structure better than the non-parametric methods. This is particularly beneficial in **longitudinal studies**, such as those in computational biology or recommendation systems, where the position of new data in an established embedding space is of significant interest. Nonparametric methods, while efficient, lack this capability.
> > >
> > > Furthermore, while speed is undoubtedly important, the quality of the embedding is paramount. Our work aims to achieve high-quality embeddings that serve the specific needs of the user, even if that means prioritizing aspects other than raw efficiency.
> > >
> > > We also wish to highlight that in the code we provided, **we also support non-parametric implementations** that are compatible with both CPU and GPU, and can also achieve comparable scalability as other non-parametric GPU implementations.

---

> > > ### Author Response · Authors · 2024-08-12
> > >
> > > > Lack of theoretical support for the method.
> > >
> > > > In my view, this method adds unnecessary complexity and introduces numerous steps without sufficient justification. The arguments seem to rely more on intuition than on solid theoretical backing.
> > >
> > > We’re not sure what you mean by “theoretical backing”. It’s hard to respond to criticism of this abstract. We provided theoretical backgrounds with the theoretical principles for loss functions in our rebuttal. It's important to note that many widely-used methods in this field, such as t-SNE, are based on **assumptions** rather than strictly theoretical foundations. For instance, there is **no definitive theory** that mandates the use of heavy-tailed Student-t distributions to represent probabilities in low-dimensional space, yet this assumption has proven effective in practice.
> > >
> > > > As a practitioner, I already find PACMAP challenging due to its numerous parameters, and this approach seems to further complicate the field unnecessarily. I strongly disagree with the authors' assertion that other methods have similar parameter complexity.
> > >
> > > With all due respect, we must strongly disagree with the assertion that PaCMAP presents excessive parameter complexity. PaCMAP is specifically designed to minimize the need for user intervention in parameter tuning. Its **default parameters** work fine for a wide range of datasets, providing a robust and reliable embedding without requiring the extensive tuning that is often necessary with methods like t-SNE and UMAP. These methods, while powerful, can indeed struggle [B] with preserving global structure unless carefully tuned—an issue PaCMAP addresses effectively.
> > >
> > > This statement also raises the question of **whether the reviewer has really looked into the implementation of any DR method** as well. Every ML algorithm has lots of parameters, including the DR methods. Even simple algorithms such as k-nearest neighbor methods have parameters - what is the form of the distance metric, how many neighbors, etc. The most popular t-SNE implementation, [openTSNE](https://github.com/pavlin-policar/openTSNE), provides **20** different tunable parameters. [UMAP](https://github.com/lmcinnes/umap) has **27** different tunable hyperparameters. These numbers do not account for the various schedules and automatic hyperparameter selection mechanisms embedded within these implementations. When considering the parametric versions of these methods, the number of hyperparameters increases significantly.
> > >
> > > [B] Kobak et al., "The art of using t-SNE for single-cell transcriptomics." Nature Communications 10.1 (2019): 5416.

---

> > > ### Author Response · Authors · 2024-08-12
> > >
> > > > From my experience, the methods that gain practical adoption are those that are straightforward to implement and easy for practitioners to grasp, minimizing the use of tricks and adhering to core principles. This is especially true in unsupervised learning, where evaluating model performance is difficult. While simplicity might come at the cost of slight performance trade-offs on the toy datasets often used in ML research, principled methods tend to perform better on real-world datasets encountered by practitioners.
> > >
> > > We agree. PaCMAP’s loss function is much simpler than t-SNE’s or UMAPs. It performs better in our experiments, which you agree to be of “rich variety of datasets and metrics” and “make sense” in your review.
> > >
> > > > TSNE is a prime example, consistently delivering state-of-the-art results across a wide range of datasets. Similarly, LargeVis, which extends TSNE with negative sampling, is another method that is accessible to ML practitioners.
> > >
> > > > The paper in question takes the opposite approach, which is why I believe it does not contribute value to the field of dimensionality reduction.
> > >
> > >
> > > There is clearly not much we can say to convince you.
> > >
> > > While we respect the contributions of t-SNE and LargeVis, we believe it is important to highlight that **newer methods**, such as PaCMAP, **offer significant advantages** in terms of maintaining global structure and simplifying the tuning process. It is not accurate to assert that older methods inherently provide superior results. In fact, PaCMAP's simpler loss function and ability to preserve global structure without extensive tuning have been key factors in its growing adoption.
> > >
> > > **We also reject the idea that LargeVis as a method is accessible to ML practitioners.** Its official implementation
> > > 1. has not been maintained for more than eight years from now,
> > > 2. needs to be compiled from scratch,
> > > 3. requires platform-specific handling, and
> > > 4. does not provide a user-friendly interface that’s compatible to popular ML frameworks, such as scikit-learn.
> > >
> > > On the other hand, our method instead is fully compatible with existing ML frameworks such as scikit-learn and pytorch.
> > >
> > > We also believe that parametric methods play an essential role in the field of dimensionality reduction. [C, D] have all garnered hundreds of citations and have inspired various scientific discoveries. We deeply regret your bias towards the parametric DR embeddings.
> > >
> > > [C] Van Der Maaten L. Learning a parametric embedding by preserving local structure, AISTATS 2009: 384-391.
> > > [D] Sainburg, Tim, Leland McInnes, and Timothy Q. Gentner. "Parametric UMAP embeddings for representation and semisupervised learning." Neural Computation 33.11 (2021): 2881-2907.

---

> ### Author Response · Authors · 2024-08-12
>
> We strongly disagree with the assessment of our paper. Here's our detailed reply:
>
> > The proposed method is limited to PACMAP.
>
> > The proposed ParamRepulsor and ParamPaCMAP are minor adaptations of the PACMAP algorithm, which is not among the most widely used neighbor embedding methods. In my view, focusing on enhancing the parametric versions of more popular algorithms like tSNE, UMAP, or LargeVis would have a much greater impact. As a result, the scope of this paper feels very limited.
>
> With all due respect, we don’t agree with the reviewer over the contribution of our work, as well as the analysis of the scope. In this work, we have clearly demonstrated the existence of local structure preservation gaps in the parametric NE methods, provided theoretical and empirical analysis over the potential causes, and then applied our methods to create a novel parametric NE DR algorithm. As practitioners ourselves, we believe PaCMAP is more reliable, robust to parameter tuning and simpler than the methods you listed - see [A] as well as the experiment section of this work, which you also found to be “rich” and “make sense”. It also has no parameters that need tuning per dataset, unlike the older methods. Additionally, despite being a more recent method published in 2021, PaCMAP has already garnered 319 citations, a number that compares favorably to LargeVis, which has received 503 citations despite being published five years earlier. This suggests that PaCMAP is gaining significant traction within the community.
>
> In terms of impact, we think **designing the best method** ought to have the largest impact. The focus of our work is on creating and improving methods that offer meaningful advances, and we believe that this should be the primary criterion for assessing the scope and impact of our research, **instead of what method we use as a basis**.
>
> [A] Huang, et al. "Towards a comprehensive evaluation of dimension reduction methods for transcriptomic data visualization." Communications biology 5.1 (2022): 719.

---

> ### Comment · Reviewer_bihR · 2024-08-12
>
> Thank you for your response, though I feel the tone could have been more respectful. To clarify, I’d like to briefly summarize my point.
>
> I have personally implemented and experimented with these algorithms and have also taught them to students on numerous occasions. This experience has led me to understand that PACMAP fundamentally involves three components for near, mid-near, and farther points, which is why I find it requires more tuning than other methods. Each component comes with its own sampling and weighting schedule.
>
> Moreover I would like to insist on some points that I believe are incorrect and potentially very misleading:
>
> - "They can also handle global structure better than the non-parametric methods." There’s no basis for this claim.
> - "This is particularly beneficial in longitudinal studies, such as those in computational biology or recommendation systems, where the position of new data in an established embedding space is of significant interest." I strongly believe that statistical analysis should not be conducted based on such parametric embeddings. **Neighbor-embedding methods should be approached with caution**, and I am convinced of the importance of considering different perspectives with various initializations, learning rates, and attraction/repulsion trade-offs. For this reason, I encourage students to compute multiple embeddings, whether using t-SNE, PACMAP, or any other method.
>
> Thank you for the discussion. I will not further discuss with authors.

---

> > ### Author Response · Authors · 2024-08-12
> >
> > > I have personally implemented and experimented with these algorithms and have also taught them to students on numerous occasions. This experience has led me to understand that PACMAP fundamentally involves three components for near, mid-near, and farther points, which is why I find it requires more tuning than other methods. Each component comes with its own sampling and weighting schedule.
> >
> > We respectfully emphasize that this work is entirely distinct from PaCMAP, and we believe that any **previous biases related to an existing work** should not influence the evaluation of this new work. As noted in our previous response, our approach does not involve more tuning, as reflected in our codespace. It’s important to highlight that, as shown in [A], methods like t-SNE and UMAP often require significant hyperparameter tuning. In contrast, all our experiments were conducted using default parameters across all datasets, as reported in our rebuttal.
> >
> > > "They can also handle global structure better than the non-parametric methods." There’s no basis for this claim.
> >
> > We respectfully disagree with the assertion that there is no basis for our claim. We have provided specific experimental results, particularly in Table 1 and Fig. 4 in the rebuttal PDF, as well as Table 4, 5, and Fig. 5-18 in the original submission. These results clearly demonstrate the better preservation of our method and parametric methods in general in handling global structures. We kindly hope you had the opportunity to thoroughly review our rebuttal, and we are open to further clarifying or expanding upon these findings if needed.
> >
> > > "This is particularly beneficial in longitudinal studies, such as those in computational biology or recommendation systems, where the position of new data in an established embedding space is of significant interest." I strongly believe that statistical analysis should not be conducted based on such parametric embeddings. Non-parametric methods should be approached with caution, and I am convinced of the importance of considering different perspectives with various initializations, learning rates, and attraction/repulsion trade-offs. For this reason, I encourage students to compute multiple embeddings, whether using t-SNE, PACMAP, or any other method.
> >
> > We hope the reviewer can provide more substantial arguments and evidence to support the belief that parametric embedding cannot be applied to parametric embeddings other than personal anecdotes. This is not the consensus of the scientific community. As we mentioned in the previous reply, statistical analysis has already been conducted with parameterized embedding and has inspired scientific discoveries, such as [E, F, G, H, I].
> >
> >
> >
> > [E] Ding et al. "Interpretable dimensionality reduction of single cell transcriptome data with deep generative models." Nature Communications 2018
> >
> > [F] Lopez et al. "Deep generative modeling for single-cell transcriptomics." Nature methods 2018
> >
> > [G] Kulichenko et al. "Uncertainty-driven dynamics for active learning of interatomic potentials." Nature Computational Science 3.3 (2023): 230-239.
> >
> > [H] Wahle et al. "Multimodal spatiotemporal phenotyping of human retinal organoid development." Nature Biotechnology 41.12 (2023): 1765-1775.
> >
> > [I] Islam et al. "Revealing hidden patterns in deep neural network feature space continuum via manifold learning." Nature Communications 14.1 (2023): 8506.

---

### Official Review · Reviewer_9imm · 2024-07-13

**Soundness:** 3
**Presentation:** 3
**Contribution:** 3
**Rating:** 6
**Confidence:** 2

**Summary:**

The paper tackles the problem of Dimensionality Reduction (DR). The authors flag a problem with the current existing parametric DR methods. It is shown with empirical evidence that parametric methods cannot capture all the local details. To mitigate this problem, the paper presents ParamRepulsor, a new parametric method that utilizes Hard Negative Mining. Empirical results show that ParamRepulsor performs strongly against the previously proposed baselines.

**Strengths:**

- DR problem is important as it is the main tool to visualize experiments and getting intuitions.
- The paper presents detailed experiments to support the conclusions
- The problem tackled is clear and well-motivated

**Weaknesses:**

See below

**Questions:**

- What is the computational time of ParamRepulsor compared to the other methods?
- What are the limitations of ParamRepulsor?

---

> ### Author Rebuttal · Authors · 2024-08-06
>
> We thank the reviewer for the detailed and encouraging review. Here are our detailed response to each question or concern raised by the reviewer:
>
> - **What is the computational time of ParamRepulsor compared to other methods?**
>
> We made a comparison of the computational time in Appendix Sec. F.2, and detailed results can be found in Figure 21. ParamRepulsor is faster than ParamUMAP, but slower than ParamInfoNC-t-SNE (which utilize the ParamCNE framework) on the two large dataset selected for comparison. ParamInfoNC-t-SNE converges faster at the cost of worse local and global structure preservation.
>
>
> - **What are the limitations of ParamRepulsor?**
>
> While ParamRepulsor creates embedding with better visual quality in general, it is not the fastest Parametric DR method. Although ParamRepulsor is faster than ParamUMAP, it requires a longer computational time, compared to ParamInfo-NC-t-SNE.

---

### Official Review · Reviewer_Tz9e · 2024-07-13

**Soundness:** 3
**Presentation:** 3
**Contribution:** 3
**Rating:** 6
**Confidence:** 4

**Summary:**

The paper addresses the problem of improving parametric neighborhood embedding (NE) methods -- i.e., techniques that optimize a neural network to project a higher dimensional dataset into a lower dimensional space. The main advantage of these being that they don't have to be recomputed for new samples, as it only involves a projection through the pre-trained encoder. However, they are generally inferior to the traditional NE techniques, the paper argues,   mainly because of failing to repulse negative pairs effectively. To address this issue, they propose a method that explicitly factors in an unsupervised hard negative mining cost, to achieve good balance of local and global structure preservation.

**Strengths:**

* The paper is well written generally and easy to follow. The motivation for their method is laid out clearly following the limitations of existing parametric and non parametric methods.
* The proposed mid-near sampling to identify hard negatives seems simple to implement, and has a tangible impact in reducing the false negatives.
* Good empirical performance from the proposed method supports the argument that including a negative repulsion loss improves local and global structure preservation.

**Weaknesses:**

* One high level comment is that some of the experiments feel incomplete in terms of answering the main hypothesis put forward -- why parametric methods under perform their non parametric counterparts. The paper identified an issue that most parametric methods suffer from -- lack of negative pair repulsion -- and addressed it using mid-near sampling. Does this make the proposed ParamRepulsor or P-PACMAP comparable or better than existing non parametric methods? The experiments only compare parametric methods so this question remains unanswered.
* The ablation from Figure 4 on the effect of weight on MN sampling clearly shows a preference for local structure preservation as it is increased -- how does this affect global structure? Every NE algorithm has some form of parameter to trade-off between these two properties. For a fair comparison, they should be ablated together to understand the relative sensitivity of the algorithms towards global and local structure preservation.
* MN Sampling utilizes uniform random sampling in high dimensions to identify hard negatives, and finds nearest neighbors presumably using an L2 like metric. I see some issues with these assumptions on data that is locally Euclidean but lives on some unknown curved manifold. Uniform sampling in high dimensions is challenging, and an L2 metric becomes inaccurate very quickly -- does the effect of hard negatives disappear as the dimensionality increases? Ablations on these properties will help the reader understand the limitations more clearly.

**Questions:**

Please see above

**Limitations:**

Limitations of the proposed method are not entirely clarified -- rather limitations of the field in general are discussed. I think this could be expanded as I have suggested in my comments.

---

> ### Author Rebuttal · Authors · 2024-08-06
>
> We thank the reviewer for the detailed review and suggestions. Here are our detailed response to each question or concern raised by the reviewer:
> - **Are parametric methods comparable to non-parametric methods?**
>
> See common response 1.
>
>
> - **How does the weight on MN sampling affect the global structure?**
>
> To clarify, the weight on the MN is not a parameter that controls the global-local tradeoff and should not be tuned by users in practice. Here, MN is only adjusted to demonstrate the effectiveness of hard negatives. As shown in Figure 4, while adjusting the MN sampling weight improves the local structure, the global relative ordering of the clusters is still preserved (see lines 216-218 and the caption).
> In our common response PDF, Figure 1 row 3 presents a thorough experiment on MNIST using 10-NN accuracy and random triplet preservation as local and global metrics, respectively (see lines 259 and 609 for definitions). We found that the weight on the MN does not significantly impact the global structure, indicating that this is not necessarily a trade-off.
>
> - **How do other NE algorithms react to parameters related to the global-local trade-off?**
>
> Non-parametric NE algorithms typically use the number of nearest neighbors to balance the global-local trade-off. To analyze how parametric NE algorithms react to the parameter change, we varied the number of nearest neighbors for ParamUMAP (P-UMAP) and ParamInfoNC-t-SNE (P-ItSNE) from 60, 30, 15 (default), 8, to 4, shifting the focus from global to local structure preservation. We then perform the evaluation for ParamRepulsor on these embeddings, visualized in common response Fig.1 row 1&2. Unlike the non-parametric case, both visual quality and quantitative metrics showed little variation. (Most likely due to the parameterization making the embedding less flexible.) In most cases, a smaller number of nearest neighbors did not result in improved local structure or degraded global structure.
>
>
> - **Effectiveness of Hard Negatives with Increasing Dimensionality.**
>
> Yes, L2 metrics are unreliable as global similarity measures in high-dimensional datasets, but we think the intuition is slightly different from what you described. For a neural network projector, a small L2 distance indicates that the **network perceives the inputs as similar** Selecting hard negative pairs with small L2 distance encourages the projector to differentiate between these similar samples. Our empirical results, spanning dimensionalities from 2 to 784 (see Appendix E), consistently show improvement over ParamPaCMAP, our baseline without hard negative sampling. Additionally, our findings align with existing research, which demonstrates that utilizing hard negatives is an effective strategy across various datasets and modalities, including those with much higher dimensionalities [1-4].
>
> - **What are the limitations of ParamRepulsor?**
>
> While ParamRepulsor creates embedding with better visual quality in general, it is not the fastest Parametric DR method. Although ParamRepulsor is faster than ParamUMAP, it requires a longer computational time, compared to ParamInfo-NC-t-SNE.
>
> [1] Robinson, Joshua, et al. "Contrastive learning with hard negative samples." ICLR 2021.
>
> [2] Wang, Yuyang, et al. "Improving molecular contrastive learning via faulty negative mitigation and decomposed fragment contrast." Journal of Chemical Information and Modeling 62.11 (2022): 2713-2725.
>
> [3] Liu, Minghua, et al. "Openshape: Scaling up 3d shape representation towards open-world understanding." NeurIPS 2023.
>
> [4] Radenovic, Filip, et al. "Filtering, distillation, and hard negatives for vision-language pre-training." CVPR 2023.

---

> > ### Comment · Reviewer_Tz9e · 2024-08-12
> >
> > Thanks for the responses. I will raise my score to 6. Only one comment regarding the confusion about parametric/non-parametric method formulation -- it maybe helpful to formulate the paper slightly differently instead of the current argument on introducing non-parametric --> parametric and then introducing the proposed method while this is the technical argument, i think the introduction can be built to emphasize on parametric methods only to avoid confusion.

---

> > > ### Author Response · Authors · 2024-08-13
> > >
> > > Thank you sincerely for taking the time to reassess our paper and for raising the score. We really appreciate your feedback and will revise our paper accordingly! Your input is invaluable to us.

---

### Author Rebuttal · Authors · 2024-08-06

We would like to thank reviewers for providing us with valuable feedback.
We have taken note of the concerns raised by each reviewer and addressed them in detail. Here, we provide responses to the most shared questions, as well as responses that require an additional PDF page. We then provide a detailed response to each reviewer's concern in the rebuttal.

1. **How does parametric method compare against non-parametric method? (KMqC, bihR)**

Non-parametric DR algorithms struggle to map new data directly into the DR plot, making them unsuitable for large, incrementally updated datasets. This weakness hinders their application in important DR application areas, such as in analyzing dataset for recommendation systems and transcriptomics. Parametric DR algorithms address this by creating a mapping from one space to another. Thus, parametric and non-parametric methods are not direct competitors, and parametric methods don't need to outperform non-parametric methods in every aspect, although we strive for that.

Our work identifies that parametric methods fall short in local structure preservation compared to non-parametric methods. This is the first study to highlight this difference and propose optimizations that enhance local structure preservation without compromising global structure. While parametric methods currently underperform in local structure preservation metrics, they excel in global structure preservation, both quantitatively and visually. Table 1 in the PDF shows the performance on random triplet accuracy for both methods. Figure 4 in the PDF visually compares selected datasets, demonstrating that parametric methods better preserve global structure in hierarchical and mammoth datasets.

2. **Would our changes be useful in a non-parametric setting? (bihR, QK23)**

In this paper, we address the performance discrepancy between parametric and nonparametric DR algorithms. Section 3 discusses how parametrization can lead to insufficient repulsive gradients, which our hard negative sampling aims to resolve. Since nonparametric DR algorithms do not face this issue, our focus was on improving parametric algorithms. However, for completeness, we also tested nonparametric repulsors and included the results in Table 1 in the PDF. Note that comparison with other parametric DR algorithms can be found in Appendix F.

---

### Author Response · Authors · 2024-08-11

Dear Reviewers,

As the discussion period is approaching its conclusion in two days, we would like to kindly remind you that we have addressed your comments in our rebuttal. We would greatly appreciate any additional feedback you may have before the deadline. If you have any further questions or concerns, please do not hesitate to reach out, and we will do our utmost to respond promptly.

Thank you for your time and consideration.

Best regards,
The Authors of Submission 13990

---

### Decision · Program_Chairs · 2024-09-25

**Decision:**

Accept (poster)

**Comment:**

The paper presents a well-motivated problem and introduces a decent technical contribution with new methods. It also provides thorough experimental evaluation to support the claims made. The reviews are largely positive, with the exception of Reviewer bihR, who raised several concerns. I believe the authors have addressed these concerns as comprehensively as possible. Given the positive reception from the other reviewers, I recommend a  **weak accept** for this paper.